

# NHM-SMAP: Spatially and temporally high resolution non-hydrostatic atmospheric model coupled with detailed snow process model for Greenland Ice Sheet

Masashi Niwano[1], Teruo Aoki[2, 1], Akihiro Hashimoto[1], Sumito Matoba[3], Satoru Yamaguchi[4], Tomonori Tanikawa[1], Koji Fujita[5], Akane Tsushima[6], Yoshinori Iizuka[3], Rigen Shimada[7], and Masahiro Hori[7]

[1]Meteorological Research Institute, Japan Meteorological Agency, Tsukuba, 305-0052 Japan
[2]Graduate School of Natural Science and Technology, Okayama University, Okayama, 700-8530 Japan
[3]Institute of Low Temperature Science, Hokkaido University, Sapporo, 060-0819 Japan
[4]Snow and Ice Research Center, National Research Institute for Earth Science and Disaster Resilience, Nagaoka, 940-0821 Japan
[5]Graduate School of Environmental Studies, Nagoya University, Nagoya, 464-8601 Japan
[6]Research Institute for Humanity and Nature, Kyoto, 603-8047 Japan
[7]Earth Observation Research Center, Japan Aerospace Exploration Agency, Tsukuba, 305-8505 Japan

*Correspondence to*: Masashi Niwano (mniwano@mri-jma.go.jp)

**Abstract.** To improve surface mass balance (SMB) estimates for the Greenland Ice Sheet (GrIS), we developed a 5km resolution regional climate model combining the Japan Meteorological Agency Non-Hydrostatic atmospheric Model and the Snow Metamorphism and Albedo Process model (NHM-SMAP) with an output interval of 1 h, forced by the Japanese 55year Reanalysis (JRA-55). We used in situ data to evaluate NHM-SMAP in the GrIS during the 2011–2014 mass balance years. We investigated two options for the lower boundary conditions of the atmosphere, an "off-line" configuration using snow/firn/ice albedo and surface temperature data from JRA-55 and an "on-line" configuration using values from SMAP. The on-line configuration improved model performance in simulating 2m air temperature, suggesting that the surface analysis provided by JRA-55 is inadequate for the GrIS and that SMAP results can better simulate snow/firn/ice physical conditions. It also reproduced the measured features of the GrIS climate, diurnal variations, and even a meso-scale strong wind event. In particular, it reproduced the GrIS surface melt area extent well. Sensitivity tests showed that the choice of calculation schemes for vertical water movement in snow and firn has an effect as great as 200 Gt year$^{-1}$ in the GrIS-wide accumulated SMB estimates; a scheme based on the Richards equation provided the best performance.





## 1 Introduction

In the Greenland Ice Sheet (GrIS), the second largest terrestrial ice sheet, a significant loss of ice mass

has been occurring since the early 1990s (e.g., Rignot et al., 2008; van den Broeke et al., 2009). Changes in the ice sheet mass (its mass balance, MB) are controlled by surface mass balance (SMB) and ice discharge across the grounding line (D), i.e., MB = SMB – D. The SMB component is related mainly to meteorological conditions and denotes the sum of mass fluxes towards the ice surface (precipitation) and away from it (runoff, sublimation, and evaporation). The Intergovernmental Panel

on Climate Change's Fifth Assessment Report (IPCC AR5) (Vaughan et al., 2013) pointed out that SMB has decreased and discharge has increased at almost the same rates since the early 1990s (van den Broeke et al., 2009), accounting for the accelerated mass loss (Rignot et al., 2011). However, more recently the situation has changed drastically as mass loss has continued to increase. Enderlin et al. (2014) attributed 84 % of the increase in the GrIS mass loss after 2009 to increased surface runoff,

which highlights the growing importance of SMB (see also Andersen et al., 2015; van den Broeke et al., 2016). Therefore, today, in situ measurements are of rising importance for monitoring changes in SMB as well as surface meteorological conditions.

Much effort has gone into monitoring surface weather conditions and SMB on the GrIS with in situ measurements. Steffen and Box (2001) established the Greenland Climate Network (GC-Net)

consisting of 18 surface automated weather stations (AWSs), distributed mainly in the accumulation area. Ahlstrøm et al. (2008) built another AWS network as part of the Programme for Monitoring of the Greenland Ice Sheet (PROMICE), with stations distributed mainly in the ablation area. van den Broeke et al. (2008) constructed an AWS network in the K-transect, a stake array along the 67°N parallel in the south-western GrIS. Aoki et al. (2014a) installed two AWSs, Snow Impurity and Glacial Microbe

effects on abrupt warming in the Arctic (SIGMA)-A and SIGMA-B, which are currently in operation in the northwestern GrIS. Regarding in situ SMB measurements, Machguth et al. (2016) compiled a large number of historical stake measurement data with a unified format, although the observations do not cover the entire GrIS. To fill geographic gaps, climate models have been developed that are constrained and calibrated by these in situ measurements. Once the validity of these models is confirmed on the

basis of the in situ data, output from the models can be used for analysis of ongoing environmental changes around the entire GrIS. These models also enable us to perform present and future climate simulations for the GrIS, including the effects of ice mass loss on global sea level rise (e.g., Rignot et al., 2011).

Several physically based regional climate models (RCMs) have been applied in the GrIS (e.g.,

MAR: Fettweis, 2007; RACMO2: Noël et al., 2015; Polar MM5: Box, 2013; and HIRHAM5: Langen et al., 2015) that have been found reliable in terms of reproducing current climate conditions (e.g., Fettweis, 2007; Box, 2013; Fausto et al., 2016; van den Broeke et al., 2016) and simulating realistic future climate change (e.g., Franco et al., 2013). Nevertheless, considerable discrepancies can be found among the SMB components simulated by these models (Vernon et al., 2013), and uncertainties in the

calculated SMBs are large compared to the uncertainties in ice discharge (Enderlin et al., 2014; van den Broeke et al., 2016). Regarding this situation, van den Broeke et al. (2016) pointed out that advances are imperative in two areas: improving the physics of SMB models and enhancing their horizontal



resolution. As for the first area, the authors noted that current models poorly represent the effects of snow/firn/ice darkening, vertical and horizontal flow of meltwater in firn or over ice lenses, and the effect of liquid water clouds on the surface energy balance as well as the resulting melt. Regarding the second area, the authors argued the necessity of statistical and dynamical downscaling from RCM outputs.

In the present study, we constructed a high-resolution polar RCM called Non-Hydrostatic atmospheric Model–Snow Metamorphism and Albedo Process (NHM-SMAP), composed of atmospheric and snowpack models developed by the Meteorological Research Institute, Japan. We employed the Japan Meteorological Agency (JMA)'s operational non-hydrostatic atmospheric model JMA-NHM (Saito et al., 2006), with a high horizontal resolution of 5 km, for dynamical downscaling. In general, a high-resolution non-hydrostatic atmospheric model has the advantage of simulating detailed meso-scale cloud structures, unlike a traditional hydrostatic atmospheric model. We also utilized the detailed physical snowpack model SMAP (Niwano et al., 2012, 2014), which features a physically based snow albedo model (Aoki et al., 2011) and a realistic vertical water movement scheme based on the Richards equation (Richards, 1931; Yamaguchi et al., 2012). Combining high-resolution detailed atmospheric and snow models is a computational challenge that has limited previous efforts of this type (e.g., Brun et al., 2011; Vionnet et al., 2014). The purpose of this study was to assess the performance of the NHM-SMAP polar RCM in reproducing current GrIS atmospheric and snow/firn/ice conditions by utilizing in situ measurements. The chosen study period, September 2011 to August 2014, includes the record surface melt event that occurred during summer 2012 (Nghiem et al., 2012; Tedesco et al., 2013; Hanna et al., 2014). Using the data, NHM-SMAP was evaluated from various aspects, where 1 hour interval model output data were employed. Typical output data from this kind of RCM have a temporal resolution of 6 h to 1 day (Cullather et al., 2016). Therefore, this study was an attempt to take advantage of both short-term detailed weather forecast models and long-term computationally stable climate models. The success of our attempt may make model output data from NHM-SMAP valuable for assessing not only long-term climate change in the GrIS but also detailed diurnal variations of the meteorological, snow, firn, and ice conditions in the GrIS.

The purposes of this paper are to describe the NHM-SMAP polar RCM and to demonstrate its capacity to reproduce current GrIS atmospheric and snow/firn/ice conditions by utilizing in situ measurements. Section 2 of this paper describes the NHM-SMAP model in detail, and the in situ measurement data for surface meteorology and SMB we used in this study are introduced in Sect. 3. Section 4 presents the results of our validation analysis and discusses their implications for the future direction of NHM-SMAP's applications. Finally, in Sect. 5 we summarize our conclusions.

## 2 Model descriptions

### 2.1 Atmospheric model JMA-NHM

JMA-NHM employs flux form equations in spherical curvilinear orthogonal coordinates as the governing basic equations. Saito et al. (2006) demonstrated that JMA-NHM outperforms the JMA's previous hydrostatic regional model in predictions of synoptic meteorological fields and quantitative





forecasts of precipitation. Although JMA-NHM is used mainly for operational daily weather forecasts around Japan, the model can also be used for long-term climate simulations (Murata et al., 2015). Recently, JMA-NHM was applied to support a field expedition in the GrIS (Hashimoto et al., 2017), and the model setting used on that occasion was followed in this study. A double-moment bulk cloud

microphysics scheme was used to predict both the mixing ratio and concentration of solid hydrometeors (cloud ice, snow, and graupel), and a single-moment scheme was used to predict the mixing ratio of liquid hydrometeors (cloud water and rain). In addition, ice crystal formation in the atmosphere was simulated by using an up-to-date formulation that depends on temperature. Following Hashimoto et al. (2007), we did not employ the ice-saturation adjustment scheme and the cumulus

parameterization used in the original configuration. The turbulence closure boundary layer scheme was formulated following the improved Mellor-Yamada Level 3 (Nakanishi and Niino, 2006). For atmospheric radiation, the transfer function in longwave radiation was computed by a random model developed by Goody (1952), and shortwave radiation was computed by diagnosing the transfer function following Briegleb (1992).

**2.2 Physical snowpack model SMAP**

The multi-layered physical snowpack model SMAP was developed for the seasonal snowy areas of Japan by Niwano et al. (2012, 2014). SMAP calculates the temporal evolution of broadband snow albedos in the UV-visible, near-infrared, and shortwave spectra as well as the internal physical parameters of snowpack such as temperature, density, grain size, and grain shape. Because the model

incorporates the physically based snow albedo model (PBSAM) developed by Aoki et al. (2011), it can assess effects of snow grain size and impurity concentration (black carbon and dust) on snow albedo explicitly in principle. SMAP calculates vertical water movement in snow and firn by employing the detailed Richards equation (Richards, 1931; Yamaguchi et al., 2012). SMAP is also equipped with a bucket scheme to calculate vertical water movement in snow and firn, in which liquid water exceeding

the maximum prescribed water content descends to the adjacent lower layer (Niwano et al., 2012). Because a bucket scheme is used in most existing polar RCMs (Reijmer et al., 2012), we investigated whether the Richards equation scheme improves the GrIS SMB (see Sect. 4.7).

Niwano et al. (2015) applied SMAP to the SIGMA-A site (Aoki et al., 2014b), on the northwestern GrIS, and demonstrated that when forced by the measured surface meteorological data, the model

reproduced the temporal evolution of the physical conditions in near-surface snow (Yamaguchi et al., 2014) during the record surface melt event of summer 2012 (Nghiem et al., 2012; Tedesco et al., 2013; Hanna et al., 2014). The authors modified the original model settings only for the effective thermal conductivity of snow and the surface roughness length for momentum. In this study, we started with the same model settings described by Niwano et al. (2015). Because this was the first attempt to

perform year-round regional simulations of the GrIS with SMAP, we were obliged to make adjustments for three snow/firn/ice physical processes: new snow density (density of falling snow), ice albedo, and effects of drifting snow.





### 2.2.1 New snow density

Previous studies have suggested that new snow density in the polar region exceeds 300 kg m⁻³ (Greuell
and Konzelmann, 1994; Lenaerts et al., 2012a), whereas new snow density in mid-latitudes is typically
around 100 kg m⁻³ (e.g., Niwano et al., 2012). For this study, we used the following parameterization
for new snow density developed by Lenaerts et al. (2012a) in Antarctica:

$$\rho_{\text{new}} = A + BT_{\text{sfc}} + CU_{10\text{m}}, \tag{1}$$

where $\rho_{\text{new}}$ is the new snow density (kg m⁻³), $T_{\text{sfc}}$ is the surface temperature (K), $U_{10\text{m}}$ is the 10m wind
speed (m s⁻¹), and the coefficients were set at $A = 97.5$ kg m⁻³, $B = 0.77$ kg m⁻³ K⁻¹, and $C = 4.49$ kg s
m⁻⁴. As an additional condition, the minimum and maximum values of $\rho_{\text{new}}$ were set at 300 and 350 kg
m⁻³ following Lenaerts et al. (2012a).

### 2.2.2 Ice albedo

Although the PBSAM snow albedo component in SMAP allows us to simulate snow albedo
realistically, its present version cannot be applied to an ice surface because the optically equivalent
grain size of high-density ice, an important input parameter, cannot be defined and calculated by
SMAP. In this study, we calculated the albedos of snow and firn with the PBSAM snow albedo
component, defining firn as snow with density between 400 and 830 kg m⁻³ following Cuffey and
Paterson (2010). The albedo of ice was calculated by a linear equation as a function of density and
ranged from 0.55, the typical albedo of clean firn (Cuffey and Paterson, 2010), to 0.45, taken from the
MAR model setting as explained by Alexander et al. (2014).

### 2.2.3 Effects of drifting snow

Sublimation of drifting snow is an important contributor to the GrIS SMB (Lenaerts et al., 2012b). In
SMAP, the drifting snow condition is diagnosed on the basis of a mobility index $M_{\text{O}}$, which describes
the potential for snow erosion of a given snow layer, and a driftability index $S_{\text{I}}$. Following Vionnet et al.
(2012), $M_{\text{O}}$ is calculated by

$$M_{\text{O}} = \begin{cases} 0.34(0.75d - 0.5s + 0.5) + 0.66F(\rho) & \text{for dendritic case} \\ 0.34(-0.583g_s - 0.833s + 0.833) + 0.66F(\rho) & \text{for non-dendritic case'} \end{cases} \tag{2}$$

where $d$ is dendricity, $s$ is sphericity, $\rho$ is snow density, and $g_s$ is geometric snow grain size (mm). Here
$d$ describes the remaining portion of the original snow grains in a snow layer, and $s$ is the ratio of
rounded versus angular snow grains (Brun et al., 1992). These two parameters are calculated by SMAP
as explained by Niwano et al. (2012). $F$ as an empirical function of density is written as

$$F(\rho) = [1.25 - 0.0042(max(50, \rho) - 50)]. \tag{3}$$

Using $M_{\text{O}}$, $S_{\text{I}}$ is diagnosed from the equation proposed by Guyomarc'h and Merindol (1998):



$$S_{\mathrm{I}} = -2.868 e^{-0.085U} + 1 + M_{\mathrm{O}}, \tag{4}$$

where $U$ is the 2m wind speed (m s$^{-1}$), and the value of $U$ when $S_{\mathrm{I}}$ becomes 0 indicates the threshold
wind speed $U_{\mathrm{t}}$ for the occurrence of drifting snow. Once the onset of the drifting snow condition is
simulated by SMAP, the drifting snow sublimation rate $F_{\mathrm{s}}$ (kg m$^{-2}$ s$^{-1}$) at 2 m above the surface is
calculated following Gordon et al. (2006):

$$F_{\mathrm{s}} = D \left(\frac{T_0}{T_{\mathrm{a}}}\right)^{\gamma} U_{\mathrm{t}} \rho_{\mathrm{a}} q_{\mathrm{si}} (1 - R_{\mathrm{Hi}}) \left(\frac{U}{U_{\mathrm{t}}}\right)^{E}, \tag{5}$$

where $T_{\mathrm{a}}$ is air temperature (K), $T_0$ is 273.15 K, $\rho_{\mathrm{a}}$ is air density (kg m$^{-3}$), $q_{\mathrm{si}}$ is saturation specific
humidity with respect to ice at temperature $T_{\mathrm{a}}$ (kg kg$^{-1}$), and $R_{\mathrm{Hi}}$ is relative humidity with respect to ice.
The dimensionless constants are $D = 0.0018$, $\gamma = 4$, and $E = 3.6$. Although it is ideal to calculate the
erosion of drifting snow (redistribution of near-surface snow caused by drifting snow), it was neglected
in NHM-SMAP because of computational costs. Lenaerts et al. (2012b) reported that the contribution
of drifting snow erosion to SMB is negligible on the GrIS; however, it is locally important, especially
in areas where topographic features induce strong divergence or convergence in the wind field.

**2.3 NHM-SMAP coupling simulation procedure**

**2.3.1 Model domain and ice sheet mask**

The 5km horizontal resolution JMA-NHM outputs hourly values of surface meteorological properties
including precipitation (snow and rain are discriminated internally), 2m air temperature, 2m relative
humidity with respect to water, 2m and 10m wind speed, surface pressure, downward shortwave and
longwave radiant fluxes, and cloud fraction in the calculation domain shown in Fig. 1. The model
domain consists of $450 \times 550$ horizontal grid cells, each cell characterized as land, sea, snow and ice,
or sea ice. The ice sheet mask for the GrIS was based on Bamber et al. (2001) as updated by Shimada
et al. (2016) from 2000 to 2014, including the ice sheet area minimum of summer 2012, on the basis of
MODIS satellite images. As a result, the modelled area of the GrIS and peripheral glaciers was $1.807 \times$
$10^6$ km$^2$, which agrees well with the estimate of $1.801 \pm 0.016 \times 10^6$ km$^2$ by Kargel et al. (2012). The
GrIS surface elevation was taken from Bamber et al. (2001).

**2.3.2 Dynamical downscaling of atmospheric field from reanalysis data with JMA-NHM**

We performed our high-resolution atmospheric calculation by using the dynamical downscaling
approach. The model atmosphere used by JMA-NHM in this study had a top height of about 22 km and
included 50 grid cells in the vertical direction based on terrain-following coordinates. The vertical grid
spacing increased with altitude from 40 m at the surface to 886 m at the top of the atmosphere. We
used JRA-55 (Kobayashi et al., 2015) for the upper, lower, and lateral boundary conditions of the
atmosphere. Simmons and Poli (2015) reported that the near-surface and lower-tropospheric warming
of the Arctic over the past 35 years is well reproduced by JRA-55, very much like the European Centre





for Medium-Range Weather Forecasts (ECMWF) Interim reanalysis (ERA-Interim) data (Dee et al., 2011). Surface physical properties, including albedo and temperature of land, sea, and sea ice, were taken from JRA-55 as the bottom boundary conditions of the atmosphere. As for those surface physical

properties of snow and ice, our two options were possible: it was given from JRA-55 or SMAP (see Sect. 2.3.4).

Although it is possible for JMA-NHM to perform long-term climate simulations in "climate simulation mode", where the atmosphere is initialized only at the beginning of the simulation period (Murata et al., 2015), in this study we used the "weather forecast mode", initializing the atmospheric

profile every day by referring to JRA-55. The purpose of this approach was to prevent large deviations between the JRA-55 and NHM-SMAP atmospheric fields. Therefore, every day a 30h long simulation was carried out starting from 1800 UTC of the previous day, and the model outputs of the last 24 h were employed after discarding output from the initial 6h spin-up period. This is the same procedure developed by Hashimoto et al. (2017) for daily weather forecasts for the GrIS.

### 235 2.3.3 SMAP calculation forced by results from JMA-NHM

We used SMAP, forced by the calculated surface meteorological data from the JMA-NHM, to simulate the temporal evolution of the top 30 m of snow, firn, and ice from September 2011 to August 2014. The initial snow/firn/ice physical conditions for the entire GrIS on 1 September 2011 were prepared by performing a 30year spin-up of the NHM-SMAP model following the procedure of Dumont et al.

(2014). We restricted the number of vertical model layers in the snow/firn/ice to 40 to limit computational costs. The vertical grid spacing increased from 1 cm at the surface to around 10 m at the bottom. We assumed zero heat flux at 30 m depth. For mass flux, runoff was calculated when meltwater or rain reached impermeable ice (density higher than 830 kg m$^{-3}$) and saturated the layer above the impermeable ice. A slush layer was not allowed to form, and the runoff mass was removed

from the GrIS instantaneously. When water reached 30 m depth and could not be retained, it was forced to run off immediately; however, this situation was quite rare during the study period.

Although the PBSAM component of the model allowed us to consider effects of snow impurities such as black carbon and dust explicitly, the relevant data were not available at high temporal resolution for the study period; therefore, we assumed a pure snow condition. Aoki et al. (2014b)

examined published concentrations of black carbon in near-surface snow in the GrIS and noted that most were less than several parts per billion by weight (ppbw). Reducing the albedo of snow by 0.01 requires 40 ppbw of black carbon in new snow and 10 ppbw in old melting snow (Warren and Wiscombe, 1980). We concluded that the measured concentrations of black carbon in the GrIS would not reduce albedo in snow, except possibly in old melting snow. Therefore, the pure snow assumption

is probably reasonable in the accumulation area of the GrIS. However, recent darkening of the GrIS (Shimada et al., 2016; Tedesco et al., 2016) has commanded attention. This effect is discussed in Sect. 4.4 and Sect. 4.7.



### 2.3.4 Interaction between the atmosphere and snow/firn/ice

In this study, we examined two configurations of the NHM-SMAP coupled model for the lower

boundary condition of the atmosphere, using snow/firn/ice albedo and surface temperature from JRA-55 or from SMAP (Sect. 2.3.2). The on-line configuration (SMAP) allowed us to simulate the interaction between the atmosphere and the surface whereas the off-line configuration (JRA-55) treated only the one-way supply of energy and mass from the atmosphere. Bellaire et al. (2017) has used the data obtained at GC-Net stations to demonstrate that the off-line version yields sufficiently accurate

input data for the detailed snow process model SNOWPACK (Lehning et al., 2002) to reproduce the measured near-surface snow density profiles at GC-Net stations.

### 2.3.5 Surface mass balance

Using NHM-SMAP, we calculated SMB, in meters of water equivalent (m w.e.), by the equation

$$SMB = P - SU_\mathrm{s} - SU_\mathrm{ds} - RU, \qquad (6)$$

where $P$ is precipitation, $SU_\mathrm{s}$ is sublimation or evaporation from the surface, $SU_\mathrm{ds}$ is sublimation from drifting snow particles, and $RU$ is runoff. As mentioned in Sect. 2.2.3, we neglected drifting snow erosion to reduce computational costs.

## 3 Observational data

### 3.1 Surface meteorology and surface melt area extent

To validate NHM-SMAP, we employed hourly surface meteorological data obtained with the AWSs of SIGMA (Aoki et al., 2014a; Niwano et al., 2015), GC-Net (Steffen and Box, 2001; Box and Rinke, 2003), and PROMICE (Ahlstrøm et al., 2008; van As et al., 2012), as listed in Table 1 and shown in

Fig. 2a. The properties we sought to validate were 2m air temperature, 2m water vapor pressure, surface pressure, 10m wind speed, downward shortwave and longwave radiant fluxes, snow/firn/ice surface temperatures, surface albedo, and snow surface height change. Our selection of AWSs was based on the availability of high quality data in adequate quantities during the study period and the elevation difference between the AWS site and the topographic model in NHM-SMAP (Sect. 2.3.1). To

compare the in situ measurements and the NHM-SMAP results, we used modelled data for the grid cell nearest to each AWS. Differences in elevation were not corrected in NHM-SMAP, although elevation differences greater than 200 m were not allowed. From GC-Net stations, only 2m air temperature, surface pressure, 10m wind speed, and downward shortwave radiant flux were taken. From PROMICE stations, all the properties except for surface height change were acquired, and SIGMA stations

provided all the properties. Because the sensor heights changed over time depending on accumulation and ablation, we calculated the 2m air temperature, 2m water vapor pressure, and 10m wind speed from the measurements by using the flux profile calculation module of SMAP (Niwano et al., 2012). Erroneous values were rejected after visual inspection, and temporal gaps left by the rejected data were not filled by interpolation.





For the extent of the surface melt area in the GrIS, we used the daily composite of satellite data developed by Mote (2007, 2014). This dataset, which was created from measurements by the Special Sensor Microwave Imager/Sounder (SSMIS), offers a daily record of surface and near-surface melting on the GrIS with 25km horizontal resolution. Hanna et al. (2014) utilized this dataset to evaluate recent changes in the GrIS melt area.

### 3.2 Surface mass balance

The SMB of the GrIS calculated by NHM-SMAP for the study period was evaluated by using data provided by PROMICE (Machguth et al., 2016) as well as ice core data from the SIGMA-D (Matoba et al., 2015) and SE-Dome (Iizuka et al., 2015) drilling sites (Table 2 and Fig. 2b). Most of the PROMICE stations are in the ablation area, whereas SIGMA-D and SE-Dome are in the accumulation area. Recently, SMB data from PROMICE were used for the validations of MAR (Fettweis et al., 2017) and the 1km horizontal resolution GrIS SMB product statistically downscaled from the daily output of RACMO2.3 (Noël et al., 2016). The validation sites were selected on the same basis as AWSs: data availability and an elevation difference less than 200 m between the site and the model. By employing the provided information for measurement periods at each site, the NHM-SMAP calculated SMB for each exact corresponding period were retrieved.

### 4 Model validation results and discussion

In this section we present validation results of the 5km resolution hourly NHM-SMAP output for the GrIS using in situ data obtained from September 2011 to August 2014. We include detailed information for mean error (ME; the average of the difference between simulated and observed values), root mean square error (RMSE), and coefficient of determination ($R^2$) to assess the model performance (see Table 3 and supplementary Tables S1 to S7). Sections 4.1 to 4.5 refer to hourly data from measurements and model simulations unless otherwise specified. Dates and times are expressed in UTC.

### 4.1 2m air temperature, 2m water vapor pressure, and surface pressure

The most important climatic parameter for this kind of polar RCM is 2m air temperature. Table 3 lists the model performance for 2m air temperature during the study period at each AWS depicted in Fig. 2a. Clearly, ME, RMSE, and $R^2$ for the on-line simulation were superior to those for the off-line simulation at almost all sites. Notable overestimates by the model (ME reached 6.6 °C at Summit, for example) were corrected in the on-line configuration (ME was less than 2.3 °C at all sites). These results suggest that the surface analysis provided by JRA-55 is of inadequate quality in the GrIS and that SMAP improves the results through the use of more realistic snow/firn/ice physical conditions. The following discussion focuses on results from the on-line simulation.

Figure 3a displays a year of observed and modelled 2m air temperature at SIGMA-A, from 1 September 2013 to 31 August 2014. The observed seasonal cycle was well reproduced by NHM-SMAP
($R^2$ = 0.95; Table 3); however, overestimation of the model was especially evident during winter



(November to March), when measured 2m air temperature sometimes reached below –30 °C; this characteristic was found at all sites. The scatterplot of measurements versus model simulations for the whole study period at SIGMA-A (Fig. 3b) also displays this tendency. A possible reason for this discrepancy is that JRA-55 overestimates the surface temperature. The JMA Climate Prediction

Division (CPD), which operationally develops JRA-55 data, recognizes that JRA-55 tends to overestimate winter surface air temperature in the polar region owing to inadequate treatment of energy exchanges between the atmosphere and the snow/firn/ice surface, especially under very stable atmospheric conditions, a failure that also affects the reproducibility of the surface inversion layer and results in underestimation of the lower tropospheric temperature (S. Kobayashi, personal

communication). Further investigation of this issue would require conducting further NHM-SMAP simulations forced by other reanalysis datasets like ERA-Interim, as done by Fettweis et al. (2017), which was beyond the scope of this study.

Tables S1 and S2 indicate statistics for the model performance in terms of 2m water vapor pressure and surface pressure. To summarize, $R^2$ for both parameters was acceptably high (more than 0.84), and

ME and RMSE were reasonable. Relatively large biases and RMSE as well as relatively low $R^2$ were found for 2m water vapor pressure at sites TAS_U, QAS_L, and QAS_U. This result suggests that NHM-SMAP forced by JRA-55 cannot adequately reproduce absolute water content in the southeastern GrIS. According to Hanna et al. (2006), the southeastern GrIS is characterized by high accumulation rates attributed to prevailing easterly winds, frequent cyclogenesis in and around Fram

Strait, and relatively high moisture availability when source air originates over a warm ocean. Stations TAS_U, QAS_L, and QAS_U are very close to the margin of our model domain (Fig. 1). Therefore, the use of a larger model domain that includes all of Svalbard may improve model results by resolving frequent cyclone activity in and around Fram Strait. Surface pressure was well simulated by NHM-SMAP, because $R^2$ was very close to 1.0 except for Summit. The slightly larger ME and RMSE for

surface pressure found at SIGMA-B, SCO_U, QAS_L, QAS_A, and NUK_U can be attributed to relatively large elevation differences between the actual topography and the topographic model (–165, 176, 85, 104, and 85 m, respectively), as indicated in Table S2.

### 4.2 10m wind speed

Moore et al. (2016) pointed out that topographic flow distortion commonly induces high-speed low-

level winds in the southern GrIS including tip jets, barrier winds, and katabatic flows. They also noted that an atmospheric model of Greenland would need a horizontal resolution of about 15 km to characterize the impact of topography on the regional wind field and climate; however, even at this resolution, features of the wind field would be under-resolved. Therefore, we investigated the reproducibility of a strong wind event observed at the TAS_U site (Fig. 2a) during the study period,

when a maximum 10m wind speed of 46.9 m s$^{-1}$ was recorded at 1700 UTC on 27 April 2013. A comparison of measured and simulated data (Fig. 4a) shows that the 5km resolution NHM-SMAP successfully reproduced the strong wind event but underestimated its maximum wind speed by about 5 m s$^{-1}$. A comparison of measured and modelled 10m wind speeds at TAS_U during the whole study period indicates that the model tended to underestimate high wind speeds (>30 m s$^{-1}$) but



overestimated relatively low wind speeds, resulting in ME, RMSE, and $R^2$ of 2.5 m s$^{-1}$, 4.3 m s$^{-1}$, and 0.68, respectively (Fig. 4b). At other sites, absolute values for ME and RMSE were smaller than those at TAS_U, and $R^2$ ranged widely between 0.13 (SCO_U) and 0.78 (KAN_U) (Table S3).

These results confirm that it is difficult for atmospheric models to reproduce surface wind fields in the southern GrIS. This problem may be solved by updating the boundary layer scheme (Sect. 2.1) and

increasing the horizontal resolution. In addition, a simple treatment of the surface roughness length for momentum (Niwano et al., 2015) also may affect surface wind speed estimates, as suggested by Amory et al. (2015). NHM-SMAP can provide synoptic weather data during strong wind events. Figure 4c, depicting the estimated surface wind speed field at 1700 UTC on 27 April 2013, shows that strong wind speeds were simulated near the southeastern margin of the GrIS. This surface strong wind event

corresponds to the Køge Bugt Fjord katabatic flow reported by Moore et al. (2016).

**4.3 Downward shortwave and longwave radiant fluxes**

The downward shortwave and longwave radiant fluxes are important elements of the GrIS surface energy balance. During 30 June to 14 July 2012, Niwano et al. (2015) visited SIGMA-A (Fig. 2a) and witnessed the record surface melt event (Nghiem et al., 2012; Tedesco et al., 2013; Hanna et al., 2014).

They reported mainly clear sky conditions until 9 July and cloudy conditions with occasional heavy rainfall after 10 July. NHM-SMAP successfully reproduced the observed temporal evolution and diurnal variation of downward shortwave radiant flux at SIGMA-A from 1 to 15 July; however, it tended to underestimate slightly when clouds appeared (Fig. 3c). This tendency was typical during the whole study period, as shown by Fig. 3d and the ME value of –13.5 W m$^{-2}$ listed in Table S4, although

the signs of ME differ from place to place. RMSE ranged from 56.0 W m$^{-2}$ (KPC_U) to 127.3 W m$^{-2}$ (KAN_L) and was close to values reported by Ohtake et al. (2013) when the operational version of JMA-NHM was validated using hourly data from Japan, and relatively accurate RMSEs were obtained in the northern GrIS (Table S4). The underestimation in cloudy conditions may arise from causes in the cloud radiation scheme or in the reproducibility of cloud amounts and types by the model.

Although the tendencies of ME for downward shortwave radiant flux vary from place to place, ME for the downward longwave radiant flux had a similar tendency across the GrIS, ranging from –25.1 W m$^{-2}$ at SIGMA-A to –10.8 W m$^{-2}$ at KAN_M (Table S5). Underestimates of downward longwave radiant fluxes at SIGMA-A were especially great during winter (November to January when observed values reached less than about 200 W m$^{-2}$) in the record from 1 September 2013 to 31 August 2014

(Fig. 3e) and over the whole study period (Fig. 3f). This characteristic was also found at other sites. One possible reason for this discrepancy is that the parent JRA-55 underestimates lower tropospheric temperatures, especially during winter (see Sect. 4.1). In addition, uncertainty in the winter cloud amount, low-level liquid clouds (Bennartz et al., 2013), and thin clouds (Cox et al., 2014) may affect the results. Improving the model would require detailed in situ measurements of cloud amount, cloud

type, and atmospheric profiles as well as intercomparisons against satellite remote sensing data like that of Van Tricht et al. (2016). A model intercomparison like that done by Inoue et al. (2006) would also aid deeper understanding of the limitations of current polar RCMs.





### 4.4 Snow/firn/ice surface temperature and albedo

We assessed the surface energy balance of the GrIS simulated by NHM-SMAP in terms of surface
temperature and albedo. Measured and simulated snow surface temperature at SIGMA-A from 1
September 2013 to 31 August 2014 agreed well, especially from May to October; however,
overestimates were obvious at temperatures below about –20 °C (Fig. 3g), much like the pattern for 2m
temperature (Sect. 4.1). As listed in Table S6, the model overestimated surface temperature at all sites
except NUK_U, where 2m temperature was also underestimated (Table 3). Therefore, the temporal
evolution of simulated surface and 2m temperatures followed the same pattern. Both ME and RMSE
for surface temperature were slightly larger than those for 2m temperature (Table 3); however, they are
reasonable because they were almost the same as those obtained in Japan (Niwano et al., 2014). It is
difficult to ascertain which physical process affected the model tendency because that would require us
to investigate the complicated atmosphere–snow/firn/ice coupled system simulated by NHM-SMAP.
One possible cause of the model's overestimation of surface temperature is overestimation of the near-
surface snow density profile, which would increase the conductive heat flux to the surface (see Sect.
4.5). For deeper insight, each physical scheme related to this problem should be investigated by stand-
alone tests utilizing detailed in situ measurements.

NHM-SMAP could not adequately reproduce surface albedo. The model tended to overestimate
surface albedo, especially in the ablation area (Fig. 5a). Similarly, the RMSE increased at lower surface
elevations (Fig. 5b). The model performance was best at SIGMA-A, in the accumulation area, and
worst at QAS_L in the ablation area, the most southerly station in this study (Table S7). ME and
RMSE at these two stations during months of the study period when the sun appeared (Fig. 5c and 5d)
show that model performance was uniformly good at SIGMA-A, covered with snow throughout the
year, but both ME and RMSE suddenly increased after June at QAS_L. These results imply that our
version of NHM-SMAP has difficulty simulating high-density firn and ice. Alexander et al. (2014) and
Fettweis et al. (2017) reported that this is also the case for the MAR model. Tedesco et al. (2016)
argued that the discrepancy between measured firn/ice albedo trends and trends modelled by MAR can
be explained by the absence in MAR of processes associated with light-absorbing impurities. The dark
microbe-rich sediment called cryoconite significantly reduces the surface albedo in the ablation area
(Takeuchi et al., 2014; Shimada et al., 2016). Therefore, future models should consider this process as
well as the possibility that NHM-SMAP overestimates snowfall during the summer period. In any case,
it is necessary to conduct in situ measurements in the ablation area to confirm what is happening in
reality.

**4.5 Snow surface height**

If a polar RCM can calculate changes in surface height realistically, it can be used to partition volume
changes supported by satellite altimetry observations into mass changes related to SMB and ice
dynamics (Kuipers Munneke et al., 2015). Therefore, we compared the modelled changes in hourly
snow surface height against in situ measurements obtained at SIGMA-A and SIGMA-B. Because the
SIGMA AWSs started operation in the summer of 2012 (Aoki et al., 2014a), comparisons were
performed for the 2012–2013 and 2013–2014 mass balance years (September to August). On the whole,



the model captured the trend of measured changes, but underestimations were apparent for both sites and years (Fig. 6). At SIGMA-A, ME and RMSE were respectively –0.19 and 0.21 m for 2012–2013 and –0.13 and 0.17 m for 2013–2014. At SIGMA-B, ME and RMSE were –0.24 and 0.26 m for 2012–
2013 and –0.04 and 0.12 m for 2013–2014. These scores are still acceptable by comparison to the SMAP validation results for seasonal snowpack in Japan (Niwano et al., 2014). As discussed in Sect. 4.7, SMB at the SIGMA-D site, located near SIGMA-A and SIGMA-B, is well reproduced by the model. Therefore, the underestimation can be attributed mainly to overestimation of simulated snow density, as mentioned in Sect. 4.4. Schemes for new snow density and the viscosity coefficient of snow
in the polar region may need to be upgraded by performing detailed laboratory experiments.

### 4.6 Melt area extent

The area of surface melt in the GrIS was extensive in the summer of 2012, setting a new record on 12 July 2012 (Nghiem et al., 2012; Tedesco et al., 2013; Hanna et al., 2014). At present, the melt area extent in the GrIS is commonly diagnosed from satellite data (Mote, 2007, 2014; Nghiem et al., 2012;
Hall et al., 2013). A polar RCM that can simulate the melt area extent realistically would enable us to investigate atmospheric and snow/firn/ice physical factors controlling the melt area extent within the same RCM framework, as was done by Fettweis et al. (2011). We compared the simulated daily melt area extent with the data of Mote (2007, 2014) during 2012 and 2013.

The daily melt area extent simulated by NHM-SMAP was diagnosed from hourly snow/firn/ice
surface temperature data and water content profiles. First, the daily maximum surface temperature was extracted at each grid point. If the value reached 0 °C and the top model layer contained water at the time when the maximum surface temperature was recorded, we considered the grid point to have experienced surface melt. Figure 7 shows that the simulated results matched the data well ($R^2$ was 0.97 and 0.94 for 2012 and 2013, respectively), and NHM-SMAP successfully reproduced the record melt
event around 12 July 2012, at which time the simulated melt area extent reached 92.4 %. The following year was relatively cold, as suggested by the maximum observed melt area extent of 44 %, and the model successfully replicated the satellite-derived results. It appears that NHM-SMAP can reliably and consistently simulate surface melt in the GrIS.

### 4.7 Surface mass balance

We evaluated the simulated SMB for the GrIS by using the PROMICE stake measurements and the ice core data obtained at SIGMA-D and SE-Dome (Table 2 and Fig. 2b). During the study period, 55 measurements were available. The basic geographic patterns of accumulation and ablation simulated for the 2011–2012, 2012–2013, and 2013–2014 mass balance years (Fig. S1) were almost the same as the annual mean SMB map created by RACMO2.3 (Noël et al., 2016).

The default version of NHM-SMAP employs the Richards equation to calculate vertical water movement in snow and firn. However, most polar RCMs employ a simpler scheme in which the maximum amount of water retained against gravity (irreducible water content) controls the vertical water movement (Reijmer et al., 2012). The irreducible water content is typically set at 2 % or 6 % of the pore volume, depending on the chosen modelling strategy. The lower of these values can induce



more rapid transport of water towards lower layers, mimicking the piping process. To examine the adequacy of the Richards equation for GrIS SMB estimates, we performed sensitivity tests in which the Richards equation scheme was replaced by bucket schemes with irreducible water contents of 2 % and 6 %. The tests employed only the stand-alone SMAP simulations forced by the atmospheric field calculated by the on-line version of NHM-SMAP, which implies that interaction between the

atmosphere and the snow/firn/ice was not considered. In the accumulation area where the observed SMB was positive, the simulated SMB agreed well with measurements during the study period regardless of the choice of vertical water movement scheme; however, the model did not capture large mass losses in which observed SMB reached values lower than –4 m water equivalent (m w.e.). The model tended to overestimate SMB in the lower part of the ablation area. In the default simulation, ME,

RMSE, and $R^2$ were 0.75 m w.e., 1.07 m w.e., and 0.86, respectively. With the bucket scheme, these scores worsened slightly, to 0.82 m w.e., 1.12 m w.e., and 0.85 for the case of 6 % irreducible water content and to 0.95 m w.e., 1.26 m w.e., and 0.85 for the case of 2 % irreducible water content. The Richards equation generally allows more water retention than the bucket scheme (Yamaguchi et al., 2012), which may result in higher near-surface density. In turn, more impermeable ice can form near

the surface and induce runoff from the near-surface layer. On the other hand, lower irreducible water content forces rapid transport of water towards lower layers as expected, which acts to prevent the formation of ice layers and thus surface mass loss.

    Although the Richards equation scheme contributed to improved SMB estimates by NHM-SMAP, the model still produced significant overestimates, especially in the ablation area. Deviations between

the measurements and the default model simulation results became larger where the measured SMB was smaller. A possible cause is overestimation of surface albedo by NHM-SMAP, especially in the ablation area (Sect. 4.4). In addition, it is possible that even at 5km resolution, NHM-SMAP cannot resolve the complex topography in the ablation area. Recently, Noël et al. (2016) demonstrated that statistical downscaling of individual SMB components from 11km resolution RACMO2.3 to a 1km ice

mask and topography (Howat et al., 2014) can improve SMB estimates owing to the correction of modelled surface elevations. It appears that statistical downscaling or further dynamical downscaling is inevitable to obtain more realistic SMB estimates. Moreover, it is imperative that we develop a realistic albedo model for high-density firn and ice that incorporates the effects of cryoconite.

    Using the SMB estimates from NHM-SMAP, we calculated the temporal evolution of accumulated

SMB over the entire GrIS during the 2011–2012, 2012–2013, and 2013–2014 mass balance years. We set the area of the GrIS and peripheral glaciers at $1.807 \times 10^6$ km$^2$, as explained in Sect. 2.3.1. The 2011–2012 and 2012–2013 mass balance years present a strong contrast as warm and cold years, respectively. van den Broeke et al. (2016) reported that in estimates by RACMO2.3, SMB for the GrIS reached its lowest value since 1958 in 2012, then increased greatly in 2013 and decreased slightly in

2014. Our model produced a similar sequence in those years, with accumulated SMBs at the end of each mass balance year of –23, 420, and 312 Gt year$^{-1}$, respectively (Fig. 9a). In each of these years, the differences in these estimates emerged after the beginning of June.

    Figures 9b to 9e show the accumulated totals of each SMB component in Eq. (6) for the same three mass balance years. They make it clear that the differences in the yearly estimates can be attributed





almost entirely to the differences in runoff amounts (Fig. 9c), the differences in $P$, $SU_s$, and $SU_{ds}$ being relatively small. As mentioned, NHM-SMAP overestimated SMB especially in the ablation area, which implies that the runoff amount is still underestimated. Future studies should upgrade the model physics in the ways mentioned above, then clarify how much the current version overestimates SMB across the entire GrIS. At the same time, it is imperative to validate the simulations of each SMB component in Eq. (6). In a comparison of SMB components from four reanalysis datasets and the MAR model, Cullather et al. (2016) found that large variations exist for all of the SMB components.

In light of the importance of runoff amount for our SMB estimates, we again investigated the sensitivity of our SMB simulations to the three different vertical water movement schemes. The results clearly showed that the vertical water movement scheme made a notable difference in our GrIS-wide SMB estimates: for the relatively warm 2011–2012 mass balance year, the accumulated SMBs were – 23, 113, and 174 Gt year$^{-1}$ for the default setting and the bucket schemes with irreducible water contents of 6 % and 2 %, respectively (Fig. 10a). Even in the other two relatively cold years, the SMB estimates deviated by as much as 100 Gt year$^{-1}$ (Figs. 10b and 10c). Clearly, the percolation and retention of water in snow and firn plays an important role in estimates of the present-day SMB for the GrIS.

**5 Summary and conclusions**

We developed the NHM-SMAP polar RCM, with 5km resolution and hourly output, to reduce uncertainties in SMB estimates for the GrIS. Combining JMA's operational non-hydrostatic atmospheric model JMA-NHM and the multi-layered physical snowpack model SMAP, it is an attempt to take advantage of both short-term detailed weather forecast models and long-term computationally stable climate models. Model output data from NHM-SMAP hold promise for assessing not only long-term climate change in the GrIS, but also detailed diurnal variations of meteorological, snow, firn, and ice conditions in the GrIS. We initialized the atmospheric profile every day by referring to JRA-55 (weather forecast mode) to minimize deviations between the JRA-55 and NHM-SMAP atmospheric fields, while simulating the physical states of snow/firn/ice without any initialization (climate simulation mode). The model, forced by the latest Japanese reanalysis data JRA-55, was evaluated in the GrIS during the 2011–2014 mass balance years using in situ data from the SIGMA, GC-Net, and PROMICE AWS networks, PROMICE SMB data, and ice core data from SIGMA-D and SE-Dome. After updating SMAP by incorporating physical processes for new (polar) snow density, ice albedo, and effects of drifting snow, we validated NHM-SMAP in terms of hourly 2m air temperature, 2m water vapor pressure, surface pressure, 10m wind speed, downward shortwave and longwave radiant fluxes, snow/firn/ice surface temperature and albedo, surface height change, daily melt area extent, and the GrIS accumulated SMB.

We first tested two options for the lower boundary conditions of the atmosphere. The off-line configuration used values for snow/firn/ice albedo and surface temperature from JRA-55, and the on-line configuration used values from SMAP calculations. The on-line version improved the model performance for 2m air temperature, suggesting that the surface analysis provided by JRA-55 is of



inadequate quality, at least for the GrIS, and that SMAP simulates more realistic snow/firn/ice physical conditions. Therefore, we continued our investigation using only the on-line version of NHM-SMAP.

Although the on-line version of NHM-SMAP reproduced a realistic history of 2m air temperature, it produced slight overestimates, especially during winter. A possible cause is overestimation by JRA-55 of surface temperatures in the parent data. JRA-55 overestimates surface air temperature in the polar region and underestimates lower tropospheric air temperature, apparently from deficient treatment of energy exchanges between the atmosphere and the snow/firn/ice surface, especially under very stable

atmospheric conditions. To confirm this reasoning would require NHM-SMAP simulations forced by other reanalysis datasets. Regarding 2m water vapor pressure, NHM-SMAP did not adequately reproduce absolute water content in the southeastern GrIS, and expanding the model domain to include all of Svalbard, where frequent cyclogenesis accompanies prevailing easterly winds, might improve this result. Surface pressure was simulated realistically. As for 10m wind speed, NHM-SMAP

successfully reproduced a Køge Bugt Fjord katabatic flow event observed at station TAS_U on 27 April 2013. Downward shortwave and longwave radiant fluxes, which are important contributors for the GrIS surface energy balance, were also reproduced adequately. Although our RMSEs for downward shortwave radiant flux were almost the same as those reported for Japan with the operational version of JMA-NHM, NHM-SMAP produced greater underestimates when clouds were

present. Possible causes for the error include the cloud radiation scheme and the reproducibility of cloud amount and cloud type. For downward longwave radiant flux, the model produced underestimates, especially during winter (November to January). A possible reason is underestimation of lower tropospheric temperature (especially during winter) by JRA-55, and results may also be affected by inadequate reproducibility of the winter cloud amount, low-level liquid clouds, and thin

clouds. Detailed in situ measurements for cloud amount, type, and atmospheric profiles would be required to improve model performance for downward radiant fluxes.

We assessed the simulated surface energy balance in the GrIS in terms of surface temperature and albedo. The model generally overestimated surface temperatures of snow/firn/ice, although our ME and RMSE values were close to those obtained in Japan. A possible cause for this overestimate is

overestimation of the near-surface density profile, as suggested by validation of snow surface height changes. The model overestimated the snow/firn/ice albedo, particularly in the ablation area, where both ME and RMSE suddenly increased after June. This finding underscores the need to develop a realistic albedo model for high-density firn and ice that allows us to consider the effects of darkening of the GrIS by cryoconite and so on. Because surface temperature and albedo were reasonably well

reproduced in the accumulation area, the model successfully simulated the GrIS melt area extent, including the record surface melt event during the warm summer of 2012 and the relatively cold year 2013.

In our assessment of the model's simulation of SMB, the ME, RMSE, and $R^2$ values during the study period were fairly good (0.75 m w.e., 1.07 m w.e., and 0.86, respectively). We performed

additional sensitivity tests in which the Richards equation scheme to calculate vertical water movement in snow and firn was replaced by simple bucket schemes with irreducible water contents of 2 % and 6 %, demonstrating that the realistic Richards equation scheme contributed to the improvement in SMB



estimates. However, the model still produced significant overestimates, especially in the ablation area. Improving this would require developing a realistic albedo model for high-density firn and ice. Moreover, statistical downscaling or further dynamical downscaling may inevitably be required to improve the SMB estimates. The estimates of accumulated SMB for the entire GrIS were also affected by the choice of vertical water movement scheme, which resulted in differences as great as 200 Gt year$^{-1}$ in our estimates. The process chosen to simulate water percolation and retention in snow and firn thus plays an important role in estimating SMB for the present-day GrIS.

## 6 Data availability

All of the NHM-SMAP model output data presented in this study are available upon request by contacting the corresponding author (Masashi Niwano, mniwano@mri-jma.go.jp).

*Author contributions.*

M. Niwano and A. Hashimoto developed the NHM-SMAP coupled system and performed numerical simulations. T. Aoki, S. Yamaguchi, K. Fujita, T. Tanikawa, S. Matoba, and Y. Iizuka contributed ideas for the model improvement. T. Aoki, S. Matoba, S. Yamaguchi, T. Tanikawa, K. Fujita, A. Tsushima, and M. Niwano prepared the SIGMA AWS data. S. Matoba and Y. Iizuka processed in situ SMB data from the SIGMA-D and SE-Dome ice cores. M. Niwano, R. Shimada, A. Hashimoto, T. Tanikawa, and M. Hori created the GrIS ice sheet mask used in this study. M. Niwano prepared the manuscript with contributions from all coauthors.

*Acknowledgements.*

We thank Tetsuhide Yamasaki for logistical and field support of our field measurements in the GrIS and Sakiko Daorana for her help during our stay in Greenland. We are grateful to Konrad Steffen (Swiss Federal Institute for Forest, Snow and Landscape Research WSL) for providing the GC-Net AWS data, Dirk van As (Geological Survey of Denmark and Greenland) for providing the PROMICE AWS and SMB data, and Thomas L. Mote as well as the National Snow & Ice Data Center for providing the satellite-derived GrIS melt area extent data. We thank Hiroshige Tsuguti, Nobuhiro Nagumo, and Syugo Hayashi of MRI for their help performing numerical calculations and post-processing with JMA-NHM with the supercomputer of MRI (Fujitsu PRIMEHPC FX100 and PRIMERGY CX2550M1). This study was supported in part by (1) the Japan Society for the Promotion of Science through Grants-in-Aid for Scientific Research number JP16H01772 (SIGMA project), JP15H01733 (SACURA project), and JP17K12817, (2) the Japan Aerospace Exploration Agency through the Global Change Observation Mission—Climate (GCOM-C)/Second-generation GLobal Imager (SGLI) Mission, (3) the Ministry of the Environment of Japan through the Experimental Research Fund for Global Environment Conservation, and (4) the Institute of Low Temperature Science, Hokkaido University, through the Grant for Joint Research Program.





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



**Table 1: Locations of observation sites for surface meteorology, including surface elevations measured on site ($z_{obs}$) and specified in NHM-SMAP ($z_{model}$).**

| Sites | Latitude (°N) | Longitude (°E) | $z_{obs}$ (m) | $z_{model}$ (m) |
|---|---|---|---|---|
| SIGMA-A | 78.05 | −67.63 | 1490 | 1494 |
| SIGMA-B | 77.52 | −69.06 | 944 | 779 |
| Summit | 72.58 | −38.51 | 3208 | 3252 |
| S-Dome | 63.15 | −44.82 | 2901 | 2921 |
| KPC_U | 79.83 | −25.17 | 870 | 893 |
| SCO_U | 72.39 | −27.24 | 980 | 1156 |
| TAS_U | 65.70 | −38.87 | 570 | 571 |
| QAS_L | 61.03 | −46.85 | 290 | 375 |
| QAS_A | 61.24 | −46.73 | 1010 | 1114 |
| NUK_L | 64.48 | −49.53 | 550 | 576 |
| NUK_U | 64.51 | −49.27 | 1130 | 1215 |
| NUK_N | 64.95 | −49.88 | 920 | 966 |
| KAN_L | 67.10 | −49.95 | 680 | 606 |
| KAN_M | 67.07 | −48.83 | 1270 | 1319 |
| KAN_U | 67.00 | −47.02 | 1840 | 1860 |
| UPE_L | 72.89 | −54.3 | 220 | 254 |
| UPE_U | 72.89 | −53.57 | 940 | 1017 |



**Table 2: Locations of observation sites for SMB, including the official ID for PROMICE sites and surface elevations measured on site ($z_{obs}$) and specified in NHM-SMAP ($z_{model}$).**


| Glacier names or sites | PROMICE ID | Latitude (°N) | Longitude (°E) | $z_{obs}$ (m) | $z_{model}$ (m) |
|---|---|---|---|---|---|
| Tuto Ramp | 120_THU_L | 76.4 | −68.26 | 570 | 576 |
| | 120_THU_U | 76.42 | −68.14 | 770 | 583 |
| Qaanaaq Ice Cap | 126_Q05 | 77.52 | −69.11 | 839 | 779 |
| Kronprins Christian Land | 170_KPC_U | 79.83 | −25.17 | 870 | 893 |
| | 220_11 | 74.66 | −21.55 | 1132 | 1270 |
| A.P. Olsen Ice Cap | 220_12 | 74.65 | −21.6 | 1226 | 1270 |
| | 220_13 | 74.66 | −21.6 | 1271 | 1270 |
| | 220_14 | 74.68 | −21.61 | 1334 | 1270 |
| Violin Glacier | 232_SCO_U | 72.39 | −27.26 | 1000 | 1156 |
| Isertoq | 270_TAS_L | 65.64 | −38.9 | 270 | 337 |
| Qassimiut Ice Lobe | 340_QAS_L | 61.03 | −46.85 | 310 | 375 |
| | 340_QAS_U | 61.18 | −46.82 | 890 | 894 |
| Qamanarssup Sermia | 414_NUK_L | 64.48 | −49.53 | 560 | 576 |
| | 414_NUK_U | 64.5 | −49.26 | 1140 | 1215 |
| Kangilinnguata Sermia | 416_NUK_N | 64.95 | −49.88 | 930 | 966 |
| | 454_S4 | 67.1 | −50.19 | 383 | 364 |
| | 454_S5 | 67.1 | −50.09 | 490 | 473 |
| | 454_SHR | 67.1 | −49.94 | 710 | 606 |
| | 454_S6 | 67.08 | −49.4 | 1010 | 1056 |
| | 454_S7 | 66.99 | −49.15 | 1110 | 1136 |
| K-Transect | 454_S8 | 67.01 | −48.88 | 1260 | 1277 |
| | 454_S9 | 67.05 | −48.25 | 1520 | 1525 |
| | 454_S10 | 67 | −47.02 | 1850 | 1860 |
| | 454_KAN_L | 67.1 | −49.93 | 680 | 606 |
| | 454_KAN_M | 67.07 | −48.82 | 1270 | 1319 |
| | 454_KAN_U | 67 | −47.02 | 1850 | 1860 |
| Upernavik | 475_UPE_L | 72.89 | −54.29 | 230 | 254 |
| | 475_UPE_M | 72.89 | −53.53 | 980 | 1017 |
| SIGMA-D | | 77.64 | −59.12 | 2100 | 2097 |
| SE-Dome | | 67.18 | −36.37 | 3170 | 3031 |



**Table 3: Model performance in simulating hourly 2m air temperature at each AWS on the GrIS (locations in Fig. 1). ME, mean error (average of the difference between simulated and observed values); RMSE, root mean square error; $R^2$, coefficient of determination.**

| Sites | Off-line configuration | | | On-line configuration | | | Number of observations |
|---|---|---|---|---|---|---|---|
| | ME (°C) | RMSE (°C) | $R^2$ | ME (°C) | RMSE (°C) | $R^2$ | |
| SIGMA-A | 2.5 | 3.7 | 0.94 | 1.5 | 3.0 | 0.95 | 18998 |
| SIGMA-B | 2.8 | 3.4 | 0.97 | 2.3 | 2.9 | 0.97 | 18540 |
| Summit | 6.6 | 8.1 | 0.88 | 2.3 | 5.2 | 0.89 | 21137 |
| S-Dome | 1.9 | 3.4 | 0.91 | 0.7 | 2.8 | 0.92 | 15059 |
| KPC_U | 3.9 | 5.5 | 0.93 | 2.3 | 4.4 | 0.94 | 26139 |
| SCO_U | 2.8 | 4.6 | 0.86 | 0.9 | 3.9 | 0.85 | 25786 |
| TAS_U | 2.8 | 3.7 | 0.84 | 2.3 | 3.2 | 0.87 | 23263 |
| QAS_L | 1.1 | 2.3 | 0.89 | 0.4 | 2.0 | 0.90 | 23483 |
| QAS_A | 0.9 | 2.8 | 0.91 | −0.3 | 2.6 | 0.92 | 8679 |
| NUK_L | 1.2 | 2.8 | 0.92 | 0.3 | 2.1 | 0.94 | 21933 |
| NUK_U | 0.4 | 2.4 | 0.93 | −0.9 | 2.4 | 0.93 | 20908 |
| NUK_N | 1.2 | 2.6 | 0.92 | 0.2 | 2.1 | 0.94 | 19955 |
| KAN_L | 2.2 | 3.3 | 0.94 | 0.9 | 2.5 | 0.95 | 25518 |
| KAN_M | 2.2 | 3.6 | 0.93 | 0.3 | 2.7 | 0.94 | 21091 |
| KAN_U | 2.6 | 4.0 | 0.94 | 0.0 | 2.7 | 0.95 | 22925 |
| UPE_L | 2.1 | 3.8 | 0.91 | 1.4 | 3.5 | 0.91 | 25434 |
| UPE_U | 1.8 | 2.9 | 0.95 | 0.4 | 2.2 | 0.96 | 23036 |





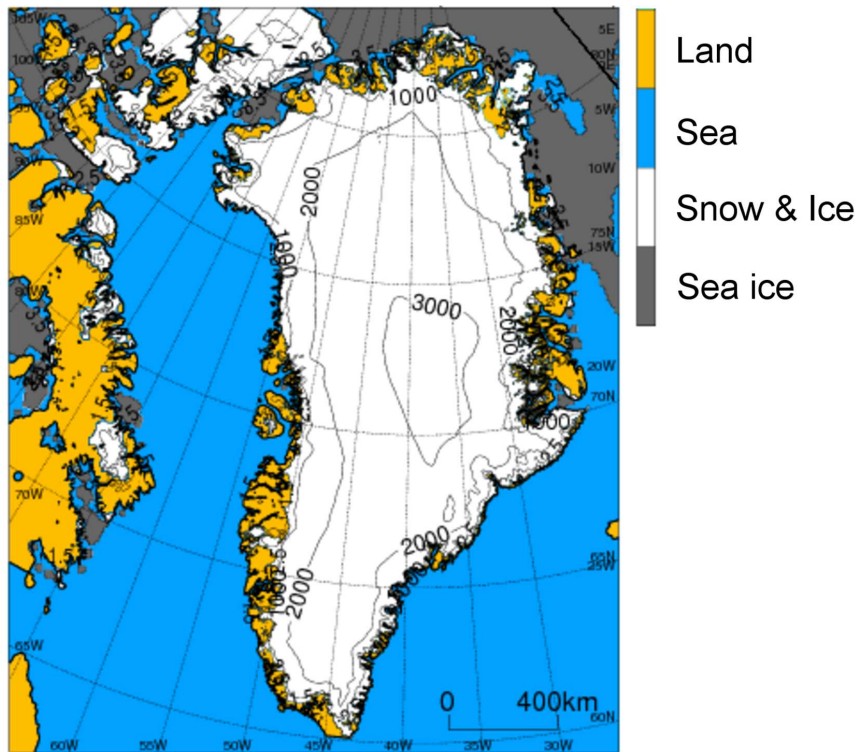

**Figure 1: Model domain of NHM-SMAP used in this study showing surface types (colours). The sea ice pattern is depicted for 1 July 2012, and it changes from day to day. Contours on ice sheets and ice caps indicate surface elevation (contour interval 1000 m).**



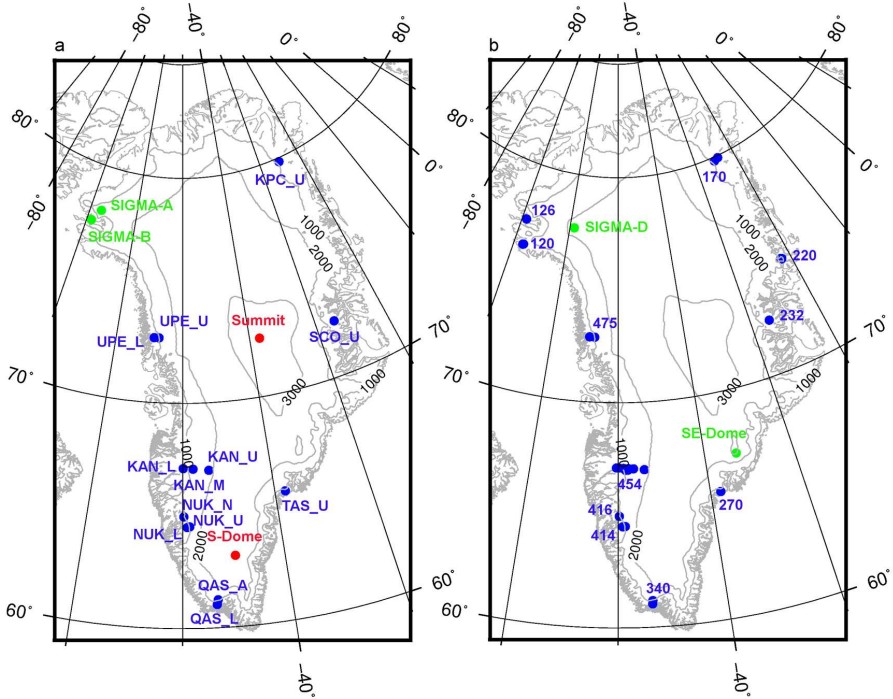

**Figure 2: Locations of observation sites for (a) surface meteorology and (b) SMB. Green circles indicate SIGMA and Japanese sites, red circles denote GC-Net sites, and blue circles represent PROMICE sites. Contours on ice sheets and ice caps indicate surface elevation (contour interval 1000 m). All sites are listed in Tables 1 and 2. Site numbers in (b) identify specific glaciers and make up the first part of the PROMICE IDs listed in Table 2.**



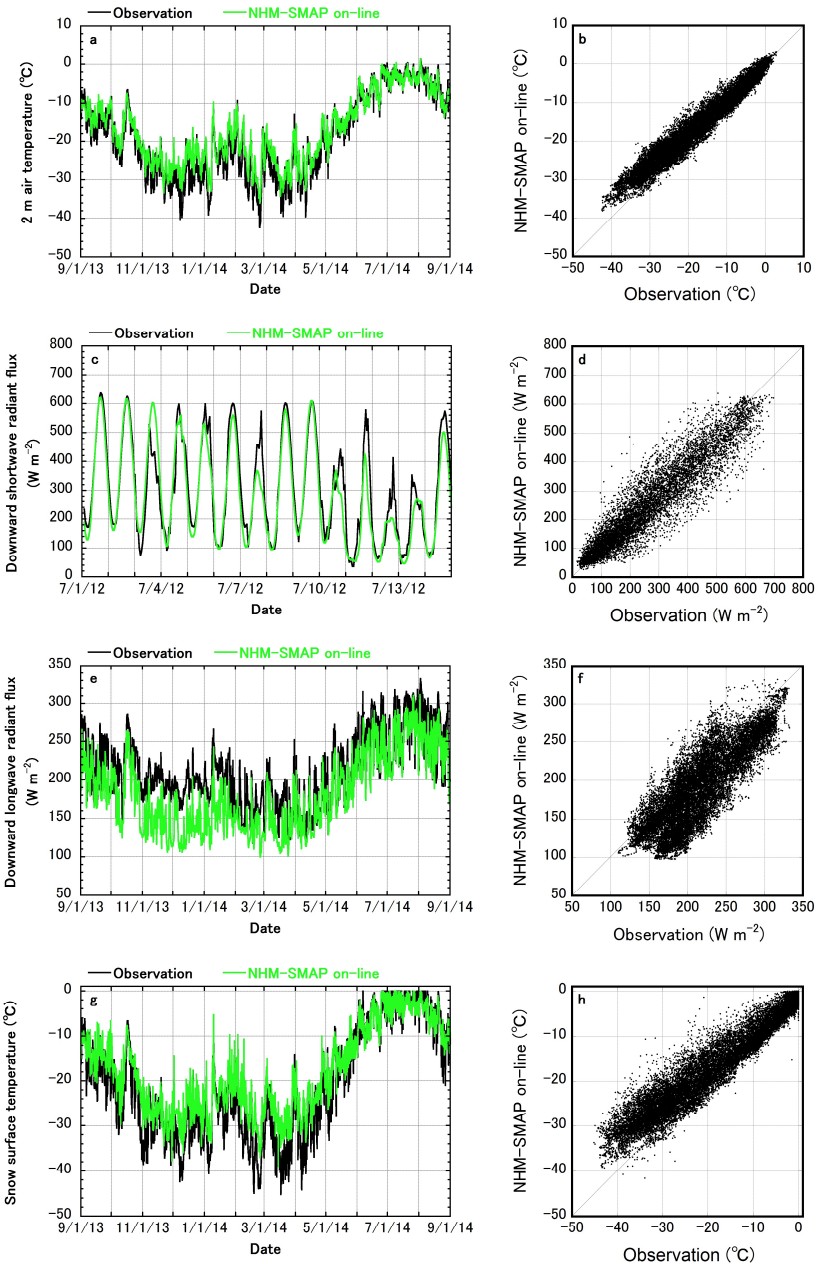

**Figure 3: Model validation of hourly (a and b) 2m air temperature, (c and d) downward shortwave radiant flux, (e and f) downward longwave radiant flux, and (g and h) snow surface temperature at SIGMA-A. Target periods for the time series on the left are (a, e, and g) 1 September 2013 to 31 August 2014 and (c) 1–14 July 2012. Data for the scatterplots on the right are from the whole study period, 1 September 2011 to 31 August 2014.**





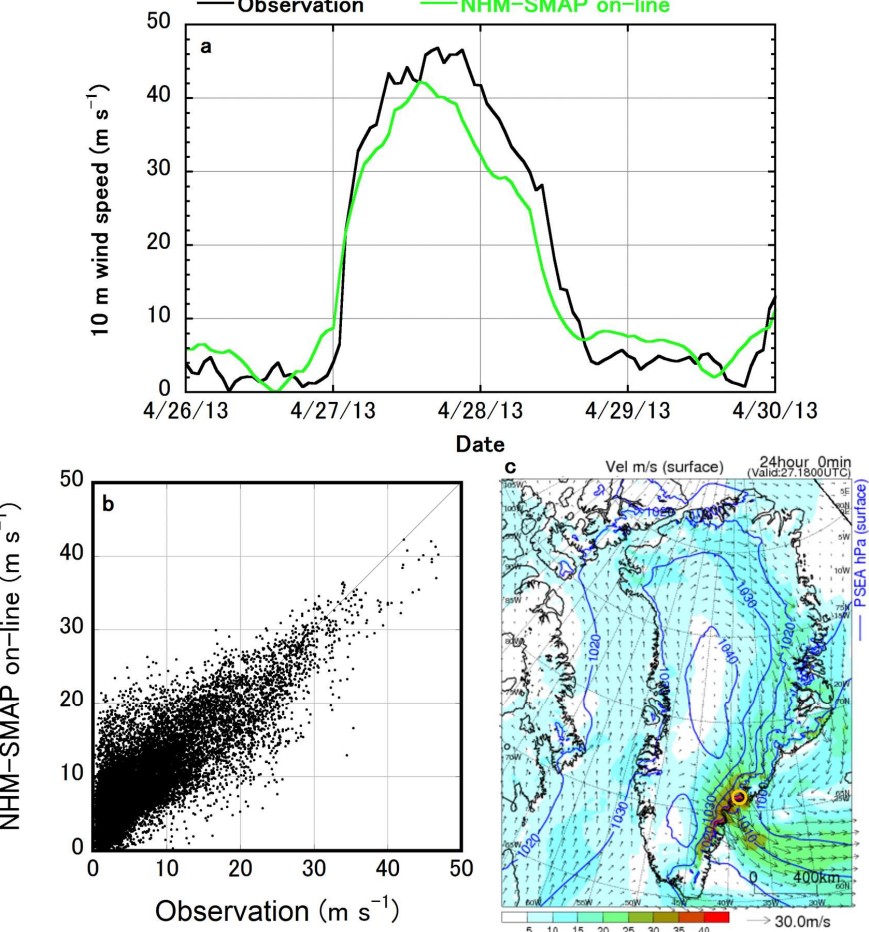

**Figure 4: Model evaluation of hourly 10m wind speed at TAS_U. (a) Time series of observed and simulated 10m wind speed at TAS_U from 26 to 29 April 2013. (b) Scatterplot of observed and simulated 10m wind speed at TAS_U during the study period. (c) Surface synoptic weather map for the model region at 1700 UTC on 27 April 2013 simulated by NHM-SMAP, showing surface wind speed (colour), surface wind vector (arrows), and sea level pressure (contours, at 10hPa**
**intervals). Open yellow circle indicates the position of TAS_U.**





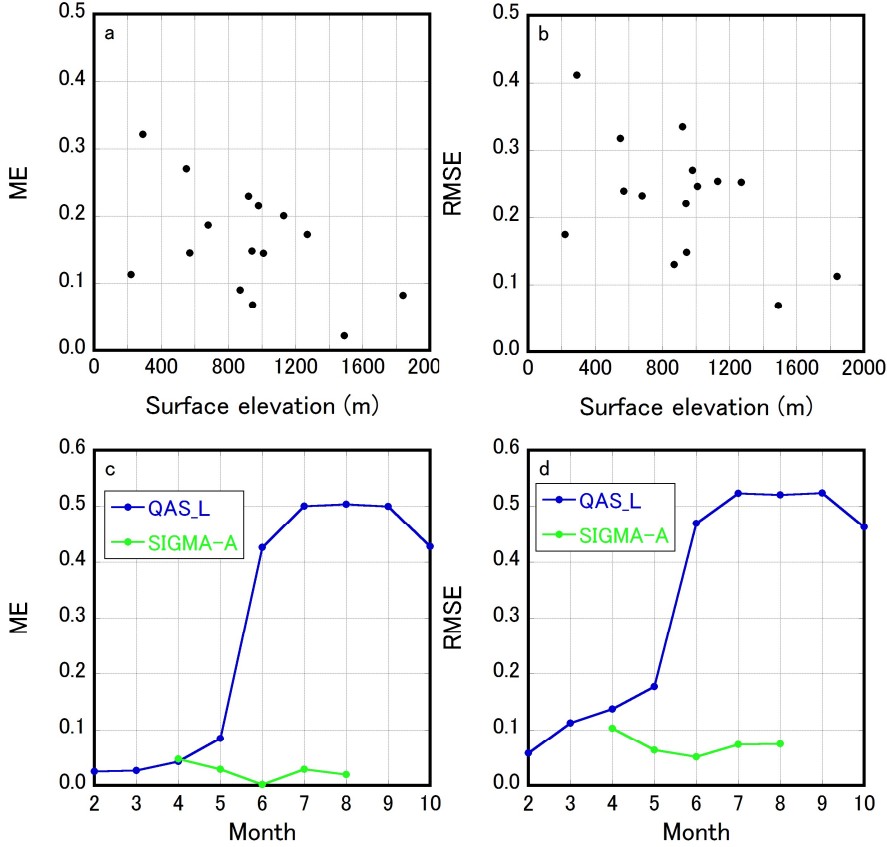

**Figure 5: Evaluation of the hourly snow/firn/ice albedo simulated at each AWS (Fig. 1 and Table S7). (a) Mean error (ME) and (b) root mean square error (RMSE) as a function of surface elevation. (c) Monthly changes in ME and (d) monthly changes in RMSE for simulated snow/firn/ice albedo at QAS_L (blue line) and SIGMA-A (green line) during months when the sun appears at each site.**






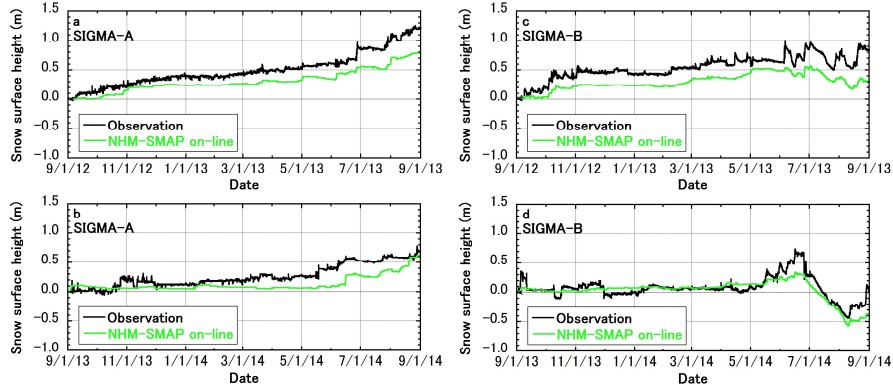

**Figure 6: Time series of observed and simulated hourly snow surface height with respect to 1 September. (a) SIGMA-A, 2012–2013; (b) SIGMA-A, 2013–2014; (c) SIGMA-B, 2012–2013; (d) SIGMA-B, 2013–2014.**





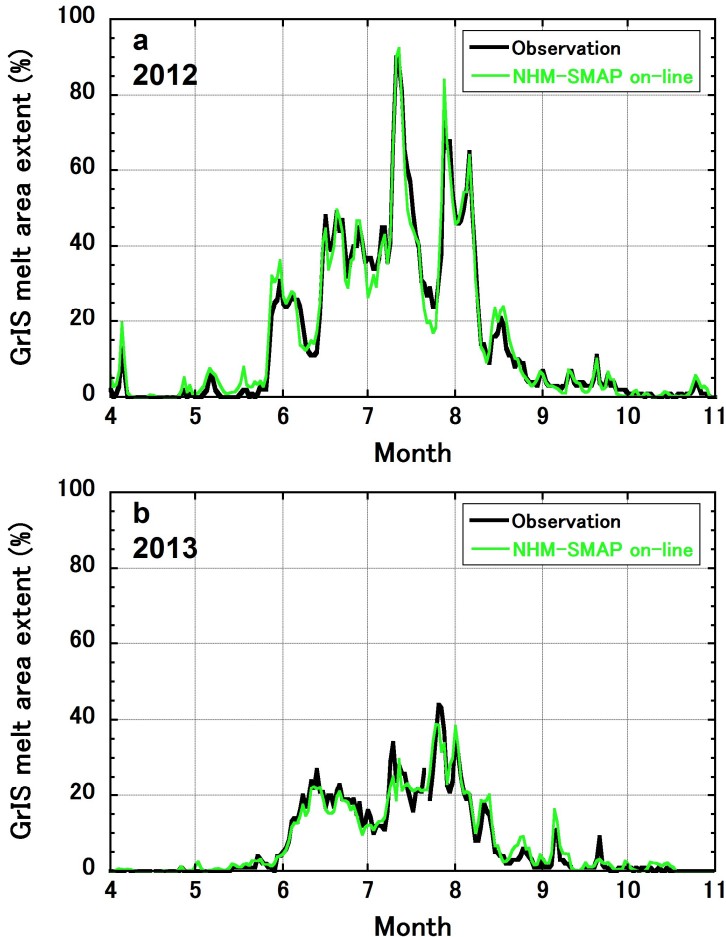

**Figure 7: Time series of observed and simulated daily GrIS melt area extent for (a) 2012 and (b) 2013. Observation data are from Mote (2014).**



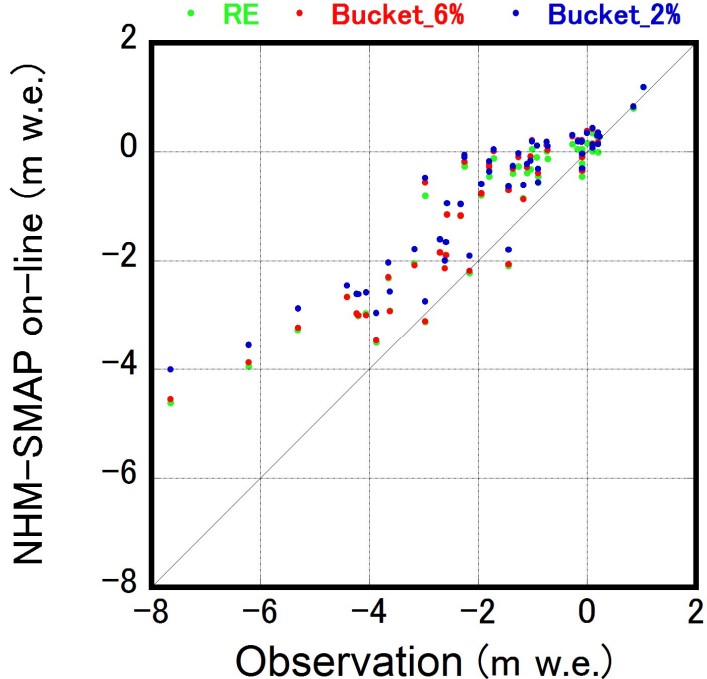

**Figure 8: Scatterplot of observed and simulated SMBs during the study period. Observation data**
**are from stake measurements compiled by PROMICE and ice core measurements from SIGMA-**
**D and SE-Dome. RE indicates the default setting for vertical water movement in snow and firn**
**based on the Richards equation; Bucket_6% and Bucket_2% are alternative settings based on**
**simple bucket schemes with irreducible water contents of 6 % and 2 % of the pore volume.**





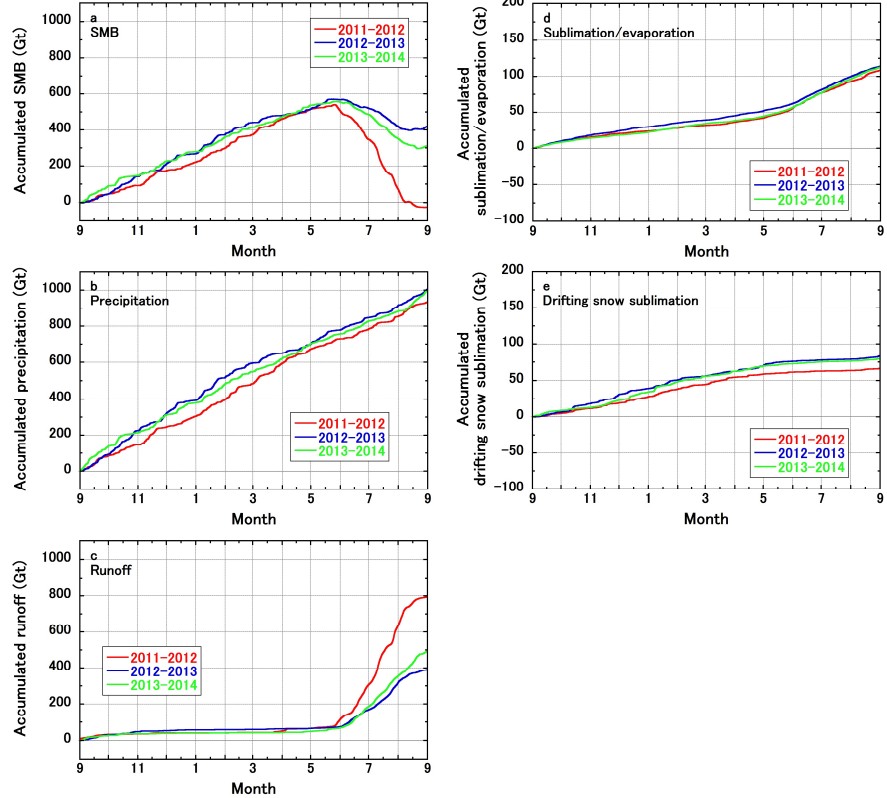


**Figure 9: Seasonal evolution of accumulated (a) SMB, (b) precipitation, (c) runoff, (d) sublimation and evaporation from the surface, and (e) drifting snow sublimation over the GrIS with respect to 1 September, during the periods 2011–2012 (red), 2012–2013 (blue), and 2013–2014 (green). Note that the vertical scale differs between the left and right columns. All results are from the default setting for vertical water movement in snow and firn based on the Richards equation.**




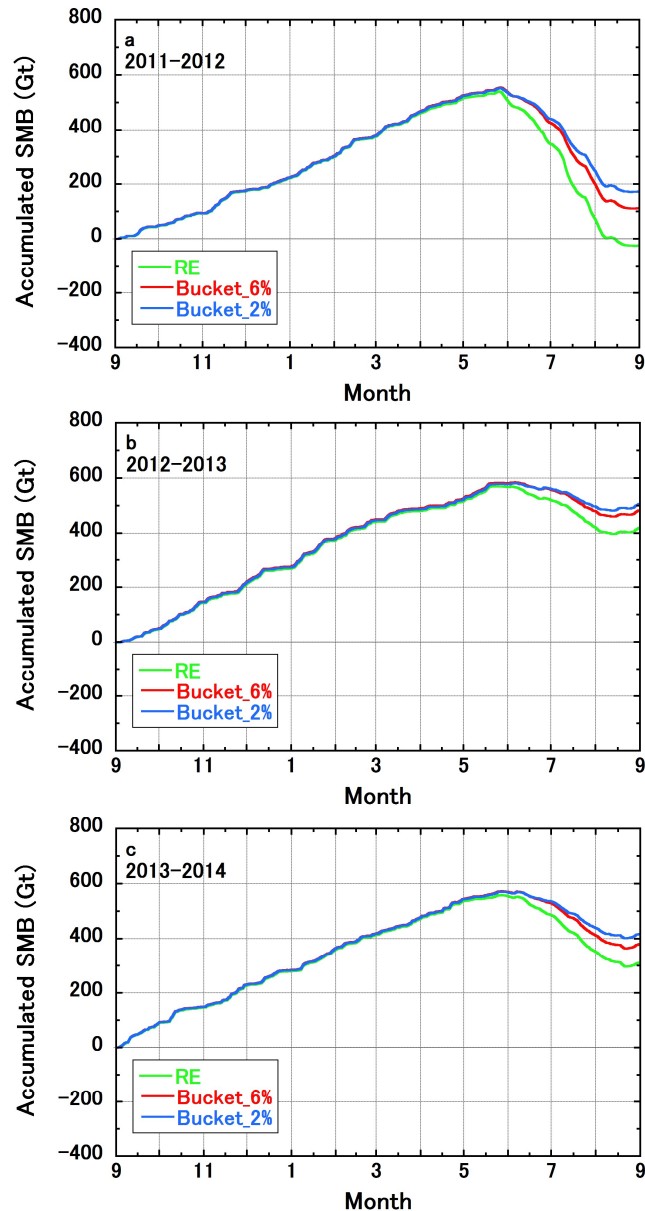

**Figure 10: Sensitivity to the choice of vertical water movement scheme of the simulated SMB for the GrIS during the (a) 2011–2012, (b) 2012–2013, and (c) 2013–2014 mass balance years. RE indicates the default setting for vertical water movement in snow and firn based on the Richards equation; Bucket_6% and Bucket_2% are alternative settings based on simple bucket schemes with irreducible water contents of 6 % and 2 % of the pore volume.**