# Peer review of "NHM-SMAP: Spatially and temporally high resolution non-hydrostatic atmospheric model coupled with detailed snow process model for Greenland Ice Sheet"

_The Cryosphere, 2017_

## Referee Comment (RC1) · Anonymous Referee #1 · 4 Jul 2017

General synopsis

This is a useful and original study of Greenland climate and surface mass balance conducted using a non-hydrostatic regional climate model. I would like to see some comparison of NHM-SMAP model output, for example as presented in Figures 9 and 10, with other RCM model output (e.g. MAR, RACMO, HIRHAM). The paper is generally well structured, written and illustrated, and should be publishable with relatively minor modifications. Citation of related work can be improved in places.

[Figure]

Specific comments

p.2, l.35 Consider adding more recent relevant references, e.g. van den Broeke 2016 The Cryosphere, Hanna et al. 2013 Nature:

van den Broeke, M. R., Enderlin, E. M., Howat, I. M., Kuipers Munneke, P., Noël, B. P. Y., van de Berg, W. J., van Meijgaard, E., and Wouters, B.: On the recent contribution of the Greenland ice sheet to sea level change, The Cryosphere, 10, 1933-1946, doi:10.5194/tc-10-1933-2016, 2016. Hanna, Edward and Navarro, Francisco J. and Pattyn, Frank and Domingues, Catia M. and Fettweis, Xavier and Ivins, Erik R. and Nicholls, Robert J. and Ritz, Catherine and Smith, Ben and Tulaczyk, Slawek and Whitehouse, Pippa L. and Jay Zwally, H. (2013) Ice-sheet mass balance and climate change. Nature, 498 (7452). pp. 51-59. ISSN: 0028-0836.

p.2, l.66: Not just RCMs but also statistically-downscaled meteorological reanalysis data have been successfully used here (Hanna et al. 2005 & 2011, Wilton et al. 2017) – please add these relevant references:

Hanna, E. and Huybrechts, P. and Janssens, I. and Cappelen, J. and Steffen, K. and Stenhens, A. (2005) Runoff and mass balance of the Greenland ice sheet: 1958-2003. Journal of Geophysical Research Atmospheres, 110 (13). ISSN: 2169-897X. Hanna, E. and Huybrechts, P. and Cappelen, J. and Steffen, K. and Bales, R. C. and Burgess, E. and McConnell, J. R. and Steffensen, J. P. and Van Den Broeke, M. and Wake, L. and Bigg, G. and Griffiths, M. and Savas, D. (2011) Greenland Ice Sheet surface mass balance 1870 to 2010 based on Twentieth Century Reanalysis, and links with global climate forcing. Journal of Geophysical Research: Atmospheres, 116 (24). ISSN: 2169-897x. Wilton, D. J. and Jowett, A. and Hanna, E. and Bigg, G. R. and Van Den Broeke, M. R. and Fettweis, X. and Huybrechts, P. (2017) High resolution (1 km) positive degree-day modelling of Greenland ice sheet surface mass balance, 1870-2012 using reanalysis data. Journal of Glaciology, 63 (237). pp. 176-193. ISSN: 0022-1430.

p.3, ll.83-85: Consider emphasising more that a key advantage of using a non-hydostatic model is its ability to be run at much higher spatial resolutions («5 km) than hydrostatic models.

Bearing the above in mind, was it considered to run the JMA-NHM at higher spatial resolutions than 5km (p.6, l.204)?

p.6, l.218 "increased with altitude from 40 m NEAR the surface to…"

p.7, l.234: "for PRODUCING daily weather forecasts…"

p.9, l.307: add that PROMICE data were also used for validating 1x1-km statistically downscaled SMB based on ERA-I reanalysis data (Wilton et al. 2017, reference as above).

p.9, l.322 "were superior on average" – quantify by how much and say whether statistically significant.

p.9, l.324: "ME was WITHIN 2.3ˆoC at all sites".

p.10, l.338: change comma to colon.

p.10, l.354: "except for Summit" – why the difference there?

p.10, l.359 add the relevant reference Orr et al. (2005): Orr, Andrew and Hanna, Edward and Hunt, Julian C. R. and Cappelen, John and Steffen, Konrad and Stephens, Ag (2005) Characteristics of stable flows over Southern Greenland. Pure and Applied Geophysics, 162 (10). pp. 1747-1778. ISSN: 0033-4553

p.14, l.11 at end of sentence suggest to add "Moreover, Wilton et al. (2017) show generally favourable results from a 1x1-km statistical downscaling of reanalysis data, with results generally comparing well with MAR and RACMO RCM output.".

p.16, l.605 after "statistical downscaling or further dynamical downscaling", add "to a higher spatial resolution than used here, e.g. 1 km (Noel et al. 2016, Wilton et al.

2017)...".

p.27, Table 3: Suggest giving mean values in new row at bottom of table.

---

## Referee Comment (RC2) · Anonymous Referee #2 · 13 Jul 2017

Review of

NHM-SMAP: Spatially and temporally high resolution non-hydrostatic atmospheric model coupled with detailed snow process model for Greenland Ice Sheet

by Niwano and others

Summary

This paper introduces a new regional climate model for use over the Greenland ice sheet. The scientific impact is modest, as a) the modelled period is relatively brief, b)

[Figure]

there clearly are issues that need to be addressed and c) the model data are not used for improved process understanding. But I presume the authors will at a later stage start using the model for these purposes. The technical quality of the figures is good, as are readability and length (apart from the last section, see below).

Major comments

l. 166: it is unclear what the physical basis is of a parameterization of ice albedo as a function of density. Ice has a near-constant density?

Section 2.2.3 explains how drifting snow sublimation at 2 m is calculated. But what is done with this information? Is a vertical sublimation profile assumed to calculate column blowing snow sublimation? Is the moisture source included in the atmospheric moisture conservation equation, i.e. is the additional water vapour used to moisten the boundary layer? What happens to surface sublimation when drifting snow sublimation starts? Please provide details to answer these questions.

l. 197: once drifting snow transport is calculated, the erosion can be simply obtained by taking the divergence of the transport. It is unclear why the authors claim that this is computationally too expensive? If it is not taken into account, the surface mass balance is locally not closed, this must at least be mentioned.

l. 210: "Ice sheet area minimum" suggests that ice sheet mask is not constant in time?

Section 3.2: How did the authors deal with the mismatch between SMB observation and model period?

Fig. 3: There is a systematic and considerable underestimation of LWin of up to 50 W m-2, which should lead to too low surface temperature, yet the snow surface temperature is overestimated in the model. I cannot reconcile this?

The summary and conclusions section can be written up much more concisely: just list the main conclusions.

Minor and textual comments

l. 167: clean firn -> clean ice

Figure 1: ice mask in Canadian Arctic looks funny.

l. 287: Why was downward lingwave radiation not used from PROMICE stations?

l. 320: Why is T2m "the most important climate parameter"? Better to leave out.

l. 473: surface melt -> surface melt extent

l. 478: "were almost the same" This is not very scientific. Please quantify or leave out. The same is true for the discussion in lines 518-520, please provide numbers.

---

## Referee Comment (RC3) · X. Fettweis (Referee) · 24 Jul 2017

This paper presents a new RCM based simulation over the Greenland ice sheet. While the scientific interest of this paper is generally poor, this "model validation" paper deserves to be published in TC and opens the door to future applications over the GrIS using a new RCM in addition to the wide commonly used RCMs family (MAR, RACMO, HIRHAM).

In addition to the justified remarks from both other reviewers, I have additional remarks

that should be resolved before publication if it is not a too big job for the authors.

pg2, line 67: site rather Fettweis et al. (2017) here

pg5, section 2.2.1: What is the sensibility of the model results to the fresh snowfall density? With MAR, the sensibility is very small and MAR uses a minimum snowfall density of 200kg/m3. 300kg/m3 is a bit high for me.

MAJOR: pg 7, line 231: As the JRA-55 surface conditions are bad (Section 4.1, line 325), is an atmospheric spin-up of 6h enough to be independent of the initial near-surface atmospheric conditions? How are the results sensitive to this spin-up time? For me, performing 48h long simulations by keeping only the last 24h will be more robust.

pg9, section 4.1: As SMAP seems to underestimates the ablation (see Fig 8), the statistics over summer (JJA) should be provided at least in supplementary material? Is the model too warm or too cold in summer ?

MAJOR: pg 10, line 341: If a RCM is totally free, it should be normally independent of the surface biases in the forcing fields. A too short spin-up time of 6h starting from too warm JRA-55 based surface conditions explains likely these biases because MARv3.5.2 forced by JRA-55 is colder in winter than MARv3.5.2 forced by ERA-Interim. Therefore, extending the spinup time should better resolve this bias than changing of forcing reanalysis. Finally, SMAP seems to underestimate LWD in winter but overestimates temperature? This is very strange?? This issue should be discussed in the paper.

pg 10, section 4.2 : I do not see the interest of showing here the ability of SMAP only to simulate a single wind event. Outputs from JRA-55 should be added in the comparison to show the interest of SMAP in respect to JRA-55. MARv3.5.2 (at a resolution of 20km) forced by JRA55 underestimates also this event by a factor of 10-15m/s. The interest of using a non-hydrostatic model at 5 km should be highlighted here.

[Figure]

pg 12, lines 409-423: the fact that SMAP overestimates surface temperature but underestimates both LWD/SWD fluxes suggests that SMAP is likely too dependent of the forcing data. What about the latent and sensible heat fluxes? The authors suggests that near-surface snow density is likely too high. I am very sceptic about this explanation. The sensibility of the results to the near-surface snow density can be tested offline. For me, the problem comes from the JRA-55 fields which are too warm and which are used every day to reinitialise the SMAP atmospheric fields.

pg 12, lines 424-439: it is true that MAR overestimates albedo but as it also overestimates SWD. Due to error compensations (as explained in Fettweis et al., 2017), the MAR surface fields are OK. Here, it is strange that SMAP overestimates temperatures but overestimates albedo and underestimates SWD and LWD.

Pg 13, section 4.6 : the comparison with the melt extent is excellent! Adding here a 2D comparison (nbr of melt days in 2012 for example) should be interesting to evaluate if this agreement is also OK locally. The simulated total melt extent could be good for bad reasons and local overestimation/underestimation of melt can be compensated.

pg 13, line 479: A 2D comparison with other RCM based estimations (RACMO, MAR, ...) is needed here for me. The raw 20km MARv3.5.2 daily outputs forced by JRA55 are available here:

ftp://ftp.climato.be/fettweis/MARv3.5.2/Greenland/JRA-55_20km/

and could be used in this paper just by citing Fettweis et al. (2017).

MAJOR: pg 14, line 507: MAR at 20km is generally able to resolve the ablation zone. The 5 km resolution used here is not an issue here to explain the systematic SMB overestimation in the ablation zone by SMAP. RACMO at 11km works also already very well. Significant biases in energy balance fluxes could explain the underestimation of ablation.

pg 14, line 513: to test the problem of the overestimation of albedo in SMAP, an offline

simulation using a bare ice albedo of 0.2 could be carried out here and results should be shown in Fig 8.

pg 14, line 522, explicit comparison with MAR or RACMO is needed here for me. RACMO or MAR time series could be added in Fig 9.

pg 15, lines 532-540: such sensitivity to the irreducible water content is also simulated by MAR which uses a value of 8%.

---

## Short Comment (SC1) · 1 Aug 2017

I agree with Xavier Fettweis that this work would be a welcome addition to the literature and the wider RCM modelling community. Some questions came up while reading the manuscript, in particular about the spinup method and the effect of percolation.

L 238-240: I searched Dumont et al. (2014) for their spin-up procedure, but failed to find information on this. Did the authors obtain the method details through personal communication?

[Figure]

L 238-240: I was wondering whether 30 years is sufficient to get a 30-m snowpack into equilibrium with the climate. Was there any remaining drift in e.g. the bottom layer temperature? What climate years were used to forced the spinup?

L 242: It reads like you started with zero snow depth at the beginning of the spinup period. The zero heat flux is then assumed at the bottom of the snow pack, not at 30 m, which is almost never reached? (which you mention in 245-246)

L484-485: Figure 10 shows that runoff is larger for larger IWC value (6%), so the "piping" effect must be dominated by something else. Otherwise, the 2%-bucket model would have produced the largest runoff value.

L 497-502: The authors do not supply any proof of their statement that the formation of ice layers is the reason for the increased runoff. In particular, they do not present melt and refreezing as separate terms. After the formation of (sub-surface) ice layers, one expects the melt to stay roughly the same order of magnitude, yet see a drop in refreezing due to the added effect of lateral runoff.

On the other hand, an increase in runoff could also occur due to increased melt. The reasoning is that when you have higher IWC and more refreezing, warmer snow will result which leads to stronger metamorphism and larger grains, that lower the albedo. The warm snow also will persist throughout winter and helps to bring snow to the melting point in spring. This behaviour is also seen in other models. It would benefit this paper if light could be shed on the exact processes that are dominant in this study.

---

## Author Comment (AC1) · 22 Sep 2017

**Reply to Reviewer#1**

We sincerely appreciate the reviewer for taking the time to provide valuable comments and suggestions. Below we describe our responses (in blue text) point-by-point to each comment (in black text). In addition, we indicate revisions in the updated manuscript together with new line numbers. Please also refer to the revised marked-up manuscript uploaded in the discussion board.

General synopsis

This is a useful and original study of Greenland climate and surface mass balance conducted using a non-hydrostatic regional climate model. I would like to see some comparison of NHM-SMAP model output, for example as presented in Figures 9 and 10, with other RCM model output (e.g. MAR, RACMO, HIRHAM). The paper is generally well structured, written and illustrated, and should be publishable with relatively minor modifications. Citation of related work can be improved in places.

We highly appreciate for this positive evaluation. In the revised manuscript, we have included simulation results from MAR v3.5.2 forced by JRA-55. At present, there are many different points in model formulations and configurations of existing RCMs, namely, resolution, ice sheet mask, dynamic core of atmospheric model, albedo model, water percolation scheme for snow/firn, etc. Therefore, detailed model inter-comparison is beyond the scope of this paper; however, we do hope to perform such a comparison in the near future. Regarding the insufficiency of references, we have included all the references suggested by the reviewer in the revised manuscript.

Specific comments

p.2, l.35 Consider adding more recent relevant references, e.g. van den Broeke 2016 The Cryosphere, Hanna et al. 2013 Nature:

van den Broeke, M. R., Enderlin, E. M., Howat, I. M., Kuipers Munneke, P., Noël, B. P. Y., van de Berg, W. J., van Meijgaard, E., and Wouters, B.: On the recent contribution of the Greenland ice sheet to sea level change, The Cryosphere, 10, 1933-1946, doi:10.5194/tc-10-1933-2016, 2016.

Hanna, Edward and Navarro, Francisco J. and Pattyn, Frank and Domingues, Catia M. and Fettweis, Xavier and Ivins, Erik R. and Nicholls, Robert J. and Ritz, Catherine and Smith, Ben and Tulaczyk, Slawek and Whitehouse, Pippa L. and Jay Zwally, H. (2013) Ice-sheet mass balance and climate change. Nature, 498 (7452). pp. 51-59. ISSN: 0028-0836.

Thank you for the suggestion. We have added these important references in the updated manuscript. (P. 2, L. 35 - 36)

p.2, l.66: Not just RCMs but also statistically-downscaled meteorological reanalysis data have been successfully used here (Hanna et al. 2005 & 2011, Wilton et al. 2017)

– please add these relevant references:

Hanna, E. and Huybrechts, P. and Janssens, I. and Cappelen, J. and Steffen, K. and Stenhens, A. (2005) Runoff and mass balance of the Greenland ice sheet: 1958-2003. Journal of Geophysical Research Atmospheres, 110 (13). ISSN: 2169-897X.

Hanna, E. and Huybrechts, P. and Cappelen, J. and Steffen, K. and Bales, R. C. and Burgess, E. and McConnell, J. R. and Steffensen, J. P. and Van Den Broeke, M. and Wake, L. and Bigg, G. and Griffiths, M. and Savas, D. (2011) Greenland Ice Sheet surface mass balance 1870 to 2010 based on Twentieth Century Reanalysis, and links with global climate forcing. Journal of Geophysical Research: Atmospheres, 116 (24). ISSN: 2169-897x.

Wilton, D. J. and Jowett, A. and Hanna, E. and Bigg, G. R. and Van Den Broeke, M. R. and Fettweis, X. and Huybrechts, P. (2017) High resolution (1 km) positive degree-day modelling of Greenland ice sheet surface mass balance, 1870-2012 using reanalysis data. Journal of Glaciology, 63 (237). pp. 176-193. ISSN:0022-1430.

We agree with this point. All the suggested papers have been listed up in the reference, and we have revised the sentence as follows:

"Several physically based regional climate models (RCMs) have been applied in the GrIS (e.g., MAR: Fettweis, 2007; RACMO2: Noël et al., 2015; Polar MM5: Box, 2013; and HIRHAM5: Langen et al., 2015) that have been found reliable in terms of reproducing current climate conditions (e.g., Fettweis, 2007; Box, 2013; Fausto et al., 2016; van den Broeke et al., 2016) and simulating realistic future climate change (e.g., Franco et al., 2013)."

----->

"Several physically based regional climate models (RCMs) (e.g., MAR: Fettweis, 2007; RACMO2: Noël et al., 2015; Polar MM5: Box, 2013; and HIRHAM5: Langen et al., 2015) and statistically-downscaled meteorological reanalysis data (Hanna et al., 2005, 2011; Wilton et al., 2017) that have been found reliable in terms of reproducing current climate conditions (e.g., Fettweis, 2007; Hanna et al., 2011; Box, 2013; Fausto et al., 2016; van den Broeke et al., 2016) have been applied in the GrIS and simulating realistic future climate change (e.g., Franco et al., 2013)." (P. 2, L. 64-69)

p.3, ll.83-85: Consider emphasising more that a key advantage of using a nonhydostatic model is its ability to be run at much higher spatial resolutions («5 km) than hydrostatic models. Bearing the above in mind, was it considered to run the JMA-NHM at higher spatial resolutions than 5km (p.6, l.204)?

Thank you for the encouraging comment. To emphasize a key advantage of a non-hydrostati model more, we have revised the sentence as follows:

"In general, a high-resolution non-hydrostatic atmospheric model has the advantage of simulating detailed meso-scale cloud structures, unlike a traditional hydrostatic atmospheric model."

----->

"In general, a non-hydrostatic atmospheric model can be run at much higher horizontal resolution (less than 10km, the limit of validity of the hydrostatic approximation) than a hydrostatic atmospheric model. Accordingly, a high-resolution non-hydrostatic atmospheric model has the advantage of simulating detailed meso-scale cloud structures, unlike a traditional hydrostatic atmospheric model. In light of recent evolution of supercomputers, it is inevitable to perform dynamical downscaling with a very high horizontal resolution, which allows us to consider effects of complex terrain like the GrIS margin on the atmospheric field explicitly." (P. 3, L. 84-90)

Regarding the latter comment, the 5km horizontal resolution was selected considering computational costs in the supercomputer of Meteorological Research Institute (Fujitsu PRIMEHPC FX100 and PRIMERGY CX2550M1). Now, the described model configuration faces a performance limit of the supercomputer. At the end of Sect. 2.3.1, we have added the following comment:

"At present, the above-mentioned domain setting faces a limitation imposed by practical computational costs in the supercomputer of Meteorological Research Institute (Fujitsu PRIMEHPC FX100 and PRIMERGY CX2550M1)." (P. 7, L. 224-226)

p.6, l.218 "increased with altitude from 40 m NEAR the surface to: : :"

OK. Revised as suggested. (P. 7, L. 238)

p.7, l.234: "for PRODUCING daily weather forecasts: : :"

The sentence has been corrected as suggested. (P. 7, L. 254)

p.9, l.307: add that PROMICE data were also used for validating 1x1-km statistically downscaled SMB based on ERA-I reanalysis data (Wilton et al. 2017, reference as above).

Thank you for the comment. We have added the explanation as follows:

"Recently, SMB data from PROMICE were used for the validations of MAR (Fettweis et al., 2017), and the 1km horizontal resolution GrIS SMB product statistically downscaled from the daily output of RACMO2.3 (Noël et al., 2016) and ERA-Interim (Wilton et al., 2017)." (P. 9, L. 332-334)

p.9, l.322 "were superior on average" – quantify by how much and say whether statistically significant.

OK. We have indicated differences in ME and RMSE from on-line and off-line simulations. In addition, significance of these differences are explained by utilizing the p-value. Now the updated sentence is as follows:
"Average ME and RMSE at all sites were improved for the on-line simulation by 1.4 °C ($p < 0.01$) and 0.7°C ($p < 0.1$), respectively." (P. 10, L. 348-349)

p.9, l.324: "ME was WITHIN 2.3ˆ°C at all sites".

Corrected as suggested. (P. 10, L. 350)

p.10, l.338: change comma to colon.

Corrected as suggested. (P. 10, L. 367)

p.10, l.354: "except for Summit" – why the difference there?

At present, we have no idea why the difference was made at Summit; however, it should be noted that ME and RMSE are still reasonable when they are compared against those obtained at other sites (Table S2). We have added the following explanation:
"Even at Summit, ME and RMSE were still reasonable when they were compared against those obtained at other sites (Table S2). The reason why R2 at Summit was relatively low should be investigated in the future." (P. 11, L. 385-387)

p.10, l.359 add the relevant reference Orr et al. (2005):
Orr, Andrew and Hanna, Edward and Hunt, Julian C. R. and Cappelen, John and Steffen, Konrad and Stephens, Ag (2005) Characteristics of stable flows over Southern Greenland. Pure and Applied Geophysics, 162 (10). pp. 1747-1778. ISSN: 0033-4553

Thank you for introducing the paper. The suggested reference has been added. (P. 11, L. 392)

p.14, l.11 at end of sentence suggest to add "Moreover, Wilton et al. (2017) show generally favourable results from a 1x1-km statistical downscaling of reanalysis data, with results generally

comparing well with MAR and RACMO RCM output.".

OK. We have added the following sentence as suggested:
"Moreover, Wilton et al. (2017) showed generally favourable results from a 1km statistical downscaling of reanalysis data, with results generally comparing well with MAR and RACMO RCM output." (P. 16, L. 580-582)

p.16, l.605 after "statistical downscaling or further dynamical downscaling", add "to a higher spatial resolution than used here, e.g. 1 km (Noel et al. 2016, Wilton et al. 2017): : :".

OK. The suggested explanation has been added. (P. 18, L. 673-675)

p.27, Table 3: Suggest giving mean values in new row at bottom of table.

Thank you for the constructive suggestion. We have added a new row indicating mean values. In addition, tables in the supplementary file has been updated in the same manner.

---

## Author Comment (AC2) · 22 Sep 2017

**Reply to Reviewer#2**

We sincerely appreciate the reviewer for taking the time to provide valuable comments and suggestions. Below we describe our responses (in blue text) point-by-point to each comment (in black text). In addition, we indicate revisions in the updated manuscript together with new line numbers. Please also refer to the revised marked-up manuscript uploaded in the discussion board.

Summary

This paper introduces a new regional climate model for use over the Greenland ice sheet. The scientific impact is modest, as a) the modelled period is relatively brief, b) there clearly are issues that need to be addressed and c) the model data are not used for improved process understanding. But I presume the authors will at a later stage start using the model for these purposes. The technical quality of the figures is good, as are readability and length (apart from the last section, see below).

Thank you for the comment. The main purpose of this paper is to present a new regional climate model for Greenland. Owing to constructive comments and suggestions provided by all the reviewers, we believe the scientific impact of the revised manuscript has been increased. Now, a long-term climate simulation by NHM-SMAP is ongoing. Obtained results will be presented in the future.

Major comments

l. 166: it is unclear what the physical basis is of a parameterization of ice albedo as a function of density. Ice has a near-constant density?

In the current model, ice albedo is set to 0.55 when surface density is 830 kg m$^{-3}$, and assumed to decrease into 0.45 that is assigned when surface density is 917 kg m$^{-3}$. The sentence has been revised as follows:

"The albedo of ice was calculated by a linear equation as a function of density and ranged from 0.55, the typical albedo of clean firn (Cuffey and Paterson, 2010), to 0.45, taken from the MAR model setting as explained by Alexander et al. (2014)."

----->

"The albedo of ice was calculated by a linear equation as a function of density and ranged from 0.55 for a surface density of 830 kg m$^{-3}$, the typical albedo of clean firn (Cuffey and Paterson, 2010), to 0.45 for a surface density of 917 kg m$^{-3}$, taken from the MAR model setting as explained by Alexander et al. (2014). " (P. 5, L. 172-175)

Section 2.2.3 explains how drifting snow sublimation at 2 m is calculated. But what is done with this information? Is a vertical sublimation profile assumed to calculate column blowing snow sublimation? Is the moisture source included in the atmospheric moisture conservation equation, i.e. is the additional water vapour used to moisten the boundary layer? What happens to surface sublimation when drifting snow sublimation starts? Please provide details to answer these questions.

Thank you for the comment. We have included the following description:
"In NHM-SMAP, surface mass loss due to drifting snow sublimation is assumed by Eq. (5); however, it is not used to moisten the boundary layer in the current version, because an interaction between the atmosphere and the snow/firn/ice surface is performed through the medium of albedo and surface temperature as mentioned later in Sect. 2.3.4." (P. 6, L. 204-208)

l. 197: once drifting snow transport is calculated, the erosion can be simply obtained by taking the divergence of the transport. It is unclear why the authors claim that this is computationally too expensive? If it is not taken into account, the surface mass balance is locally not closed, this must at least be mentioned.

We agree with reviewer that this is an important point for a model that calculates GrIS SMB. We have revised the sentence as follows:
"Although it is ideal to calculate the erosion of drifting snow (redistribution of near-surface snow caused by drifting snow), it was neglected in NHM-SMAP because of computational costs."
----->
"Although it is ideal to calculate the erosion of drifting snow (redistribution of near-surface snow caused by drifting snow), tracking changes in physical conditions of snow particles (prognostic variables of SMAP, namely, snow grain size, grain shape, density, and so on) during a drifting snow event and redistributing them in an updated surface field demands substantial computational costs. Therefore, the current version of NHM-SMAP neglects this process, which implies that simulated SMB is not closed locally." (P. 6, L. 209-214)

l. 210: "Ice sheet area minimum" suggests that ice sheet mask is not constant in time?

Our ice sheet mask is constant in time. The original description might cause misunderstanding, therefore, it has been revised as follows:

"The ice sheet mask for the GrIS was based on Bamber et al. (2001) as updated by Shimada et al. (2016) from 2000 to 2014, including the ice sheet area minimum of summer 2012, on the basis of MODIS satellite images."

----->

"The ice sheet mask for the GrIS, which is constant in time, was based on Bamber et al. (2001) as updated by Shimada et al. (2016) on the basis of 2000 to 2014 MODIS satellite images." (P. 7, L. 226-228)

Section 3.2: How did the authors deal with the mismatch between SMB observation and model period?

We referred the metadata of PROMICE SMB data and comprehended observation period. The NHM-SMAP calculated SMB data at each PROMICE site were retrieved during the exact measurement period. It is mentioned even in the original manuscript (at the end of Sect. 3.2).

Fig. 3: There is a systematic and considerable underestimation of LWin of up to 50 W m-2, which should lead to too low surface temperature, yet the snow surface temperature is overestimated in the model. I cannot reconcile this?

In the original manuscript, we mentioned possible causes for the discrepancy in terms of only insufficiencies of the model. However, we think there is also a problem in the measurement data. In the revised manuscript, we have discussed the issue as follows:

"On the other hand, observation data for downward longwave radiant flux can also have error especially during the winter period due to riming, which may act to increase measured values. In SIGMA-A, measured 2m air temperature often decreased to about –40 °C during the 2013-2014 winter (Fig. 3a). Although such reductions in 2m air temperature during March and April 2014 were followed by significant reductions in downward longwave radiant flux (Fig. 3e), they did not synchronize in December 2013 and January 2014. These results suggest that observed downward longwave radiant flux especially during December 2013 and January 2014 were affected by riming and forced to increase. A reliable quality control technique for automatic downward longwave radiant flux measurements in the polar region should be developed in the future to perform not only model validation but also climate monitoring accurately." (P. 12, L. 443 – P. 13, L. 452)

In the summary and conclusions section, an additional summary regarding this issue has been added as follows:

"On the other hand, observation data for downward longwave radiant flux can also have error especially during the winter period due to riming, which might affect the evaluation." (P. 18, L. 651-653)

During the revision, we performed additional data quality control for downward longwave radiant flux. What we performed is that rejecting such data as downward and upward longwave radiant fluxes agree exactly. This situation is caused when extreme riming occurs and these two properties are diagnosed only from sensor temperature. However, our discussion was not affected by the reassessment of measurement data.

The summary and conclusions section can be written up much more concisely: just list the main conclusions.

The first paragraph of the summary and conclusions section have been updated as follows: "We developed the NHM-SMAP polar RCM, with 5km resolution and hourly output, to reduce uncertainties in SMB estimates for the GrIS. Combining JMA's operational non-hydrostatic atmospheric model JMA-NHM and the multi-layered physical snowpack model SMAP, it is an attempt to take advantage of both short-term detailed weather forecast models and long-term computationally stable climate models. Model output data from NHM-SMAP hold promise for assessing not only long-term climate change in the GrIS, but also detailed diurnal variations of meteorological, snow, firn, and ice conditions in the GrIS. We initialized the atmospheric profile every day by referring to JRA-55 (weather forecast mode) to minimize deviations between the JRA-55 and NHM-SMAP atmospheric fields, while simulating the physical states of snow/firn/ice without any initialization (climate simulation mode). The model, forced by the latest Japanese reanalysis data JRA-55, was evaluated in the GrIS during the 2011–2014 mass balance years using in situ data from the SIGMA, GC-Net, and PROMICE AWS networks, PROMICE SMB data, and ice core data from SIGMA-D and SE-Dome. After updating SMAP by incorporating physical processes for new (polar) snow density, ice albedo, and effects of drifting snow, we validated NHM-SMAP in terms of hourly 2m air temperature, 2m water vapor pressure, surface pressure, 10m wind speed, downward shortwave and longwave radiant fluxes, snow/firn/ice surface temperature and albedo, surface height change, daily melt area extent, and the GrIS accumulated SMB."
----->
"We developed the NHM-SMAP polar RCM, with 5km resolution and hourly output, to reduce uncertainties in SMB estimates for the GrIS. Combining JMA's operational non-hydrostatic atmospheric model JMA-NHM and the multi-layered physical snowpack model SMAP, it is an

attempt to take advantage of both short-term detailed weather forecast models and long-term computationally stable climate models. The model, forced by the latest Japanese reanalysis data JRA-55, was evaluated in the GrIS during the 2011–2014 mass balance years using in situ data from the SIGMA, GC-Net, and PROMICE AWS networks, PROMICE SMB data, and ice core data from SIGMA-D and SE-Dome." (P. 17, L. 617-623)

Minor and textual comments
l. 167: clean firn -> clean ice

I checked Cuffey and Paterson (2010) again, and confirmed this description is correct. In the book, albedo for clean ice is recommended to be 0.35.

Figure 1: ice mask in Canadian Arctic looks funny.

It is true we did not examine ice mask in Canadian Arctic sufficiently, because we focus the GrIS SMB in the present study. In the revised manuscript, we have mentioned this as follows:
"In the Canadian Arctic Archipelago, considerations for details in the ice sheet mask were nod given in the present study, because we focused the GrIS SMB. Therefore, there is room for improvement on the modelled ice sheet mask, which is a future issue for NHM-SMAP." (P. 7, L. 230-233)

In connection with this point, we recognized that a resolution of Fig. 1 was not enough. Therefore, the quality of Fig. 1 has been improved in the revised manuscript.

l. 287: Why was downward longwave radiation not used from PROMICE stations?

Downward longwave radiation data from PROMICE stations are used even in the original manuscript. Model performance at each PROMICE station are indicated in Table S5. At GC-Net stations, downward longwave radiation data were not employed in the present study, because they were not measured directly during the study period.

l. 320: Why is T2m "the most important climate parameter"? Better to leave out.

Thank you for the comment. We have deleted the sentence as suggested.

l. 473: surface melt -> surface melt extent

It is an important point. We have revised as suggested. (P. 14, L. 520)

l. 478: "were almost the same" This is not very scientific. Please quantify or leave out. The same is true for the discussion in lines 518-520, please provide numbers.

Regarding the former comment, we have revised the sentence as follows:
"The basic geographic patterns of accumulation and ablation simulated for the 2011–2012, 2012–2013, and 2013–2014 mass balance years (Fig. S1) were almost the same as the annual mean SMB map created by RACMO2.3 (Noël et al., 2016)."
----->
"The geographic patterns of accumulation and ablation simulated for the 2011–2012, 2012–2013, and 2013–2014 mass balance years simulated by NHM-SMAP are depicted in Fig. S2." (P. 15, L. 527-528)

As for the latter comment, we revised the manuscript by referring to the MAR model data provided by Xavier Fettweis (Reviewer #3), and now the description has been updated as follows:
"van den Broeke et al. (2016) reported that in estimates by RACMO2.3, SMB for the GrIS reached its lowest value since 1958 in 2012, then increased greatly in 2013 and decreased slightly in 2014. Our model produced a similar sequence in those years, with accumulated SMBs at the end of each mass balance year of –23, 420, and 312 Gt year$^{-1}$, respectively (Fig. 9a)."
----->
"According to simulation results by MAR v3.5.2 forced by JRA-55 (Fettweis et al., 2017), the GrIS SMB during the 2011-2012 mass balance year was relatively low (147 Gt year$^{-1}$), then increased greatly in 2012-2013 (473 Gt year$^{-1}$) and decreased slightly in 2013-2014 (403 Gt year$^{-1}$). Our model, which tends to simulate lower SMB compared to MAR v3.5.2 that uses the bucket schemes with irreducible water contents of 8 %, produced a similar sequence in those years, with accumulated SMBs at the end of each mass balance year of –23, 420, and 312 Gt year$^{-1}$, respectively (Fig. 9a)." (P. 16, L. 591-596)

---

## Author Comment (AC3) · 22 Sep 2017

**Reply to Xavier Fettweis (Reviewer#3)**

We sincerely appreciate Xavier Fettweis for taking the time to review our paper and providing the MAR model output data as reference information. Below we describe our responses (in blue text) point-by-point to each comment (in black text). In addition, we indicate revisions in the updated manuscript together with new line numbers. Please also refer to the revised marked-up manuscript uploaded in the discussion board.

This paper presents a new RCM based simulation over the Greenland ice sheet. While the scientific interest of this paper is generally poor, this "model validation" paper deserves to be published in TC and opens the door to future applications over the GrIS using a new RCM in addition to the wide commonly used RCMs family (MAR, RACMO, HIRHAM). In addition to the justified remarks from both other reviewers, I have additional remarks that should be resolved before publication if it is not a too big job for the authors.

Thanks to insightful comments and suggestions provided by all the reviewers, we believe the manuscript has been improved and scientific quality of the revised paper has been increased.

pg2, line 67: site rather Fettweis et al. (2017) here

Revised as suggested. (P. 2, L. 67)

pg5, section 2.2.1: What is the sensibility of the model results to the fresh snowfall density? With MAR, the sensibility is very small and MAR uses a minimum snowfall density of 200kg/m3. 300kg/m3 is a bit high for me.

Thank you for the comment. In fact, NHM-SMAP's sensitivity to the fresh snowfall density has not been investigated yet. The reason why we used the parameterization by Lenaerts et al. (2012a) is simple: this is based on in-situ measurements in polar region. If smaller fresh snowfall density is set in NHM-SMAP, underestimation of snow surface height discussed in Sect. 4.5 can be solved; however, I think we don't have enough measurement-based information for fresh snowfall density to change the model scheme now.

MAJOR: pg 7, line 231: As the JRA-55 surface conditions are bad (Section 4.1, line 325), is an atmospheric spin-up of 6h enough to be independent of the initial near surface atmospheric conditions? How are the results sensitive to this spin-up time? For me, performing 48h long

simulations by keeping only the last 24h will be more robust.

Please note that insufficient conditions of JRA-55 surface analysis was unraveled through the present study. In addition, it should be noted that an appropriate spin-up period has not been established yet. An appropriate spin-up period can be found by performing a large number of simulations. The reason why we employed 6h spin-up time in the present study is that it is a typical model configuration in Japan. However, we agree with the point that further consideration of an atmospheric spin-up time can be effective to improve the model performance. The 6h spin-up period might not be suitable in the GrIS, whereas the setting seems to be effective empirically in Japan. In Sect. 4.1, we have added the following discussion:
"This result in turn suggests that making every day atmospheric spin-up period (6h; Sect. 2.3.2) longer than 6h can improve the performance of NHM-SMAP. Finding an appropriate spin-up period in the GrIS is a future issue to be coped with." (P. 10, L. 353-355)

pg9, section 4.1: As SMAP seems to underestimates the ablation (see Fig 8), the statistics over summer (JJA) should be provided at least in supplementary material? Is the model too warm or too cold in summer?

Thank you for the constructive comment. In Sect. 4.7 entitled as "Surface mass balance", we have added the following discussion:
"As presented in Sect. 4.1, the on-line version of NHM-SMAP successfully reproduced 2m air temperature at SIGMA-A during summer. Because surface mass loss during the summer is affected by near-surface (2m) temperature, model performance in terms of simulating JJA 2m air temperature at each AWS on the GrIS were re-examined (Table S8). As indicated in the table, significant or systematic error are not found, and obtained ME and RMSE are well (around –0.2 and 2.1 °C, respectively). Therefore, ---" (P. 16, L. 565-570)

MAJOR: pg 10, line 341: If a RCM is totally free, it should be normally independent of the surface biases in the forcing fields. A too short spin-up time of 6h starting from too warm JRA-55 based surface conditions explains likely these biases because MARv3.5.2 forced by JRA-55 is colder in winter than MARv3.5.2 forced by ERA-Interim. Therefore, extending the spinup time should better resolve this bias than changing of forcing reanalysis. Finally, SMAP seems to underestimate LWD in winter but overestimates temperature? This is very strange?? This issue should be discussed in the paper.

We think that a RCM cannot be totally free, because RCM-simulated atmospheric field is generally

constrained by a parent reanalysis data in lateral and upper boundaries of the RCM model domain. We imagine that simulated atmospheric field can be "almost" independent of the forcing data if we employ the "climate simulation mode", where the atmosphere is initialized only at the beginning of the simulation period, as employed by MAR. It seems to us that the present NHM-SMAP model configuration called "weather forecast mode" that initializes the atmospheric profile every day by referring to the forcing data is affected strongly by the parent data compared to the climate simulation mode. Based on this consideration, we agree with the reviewer's point that extending the spin-up time can resolve the reported bias. We have added the following discussion:

"At the same time, extending the atmospheric spin-up period discussed above can also resolve the issue, because simulation results are expected to less susceptible to a parent reanalysis data." (P. 10, L. 371 – P. 11, L. 373)

In the summary and conclusions section, it is mentioned again as follows:
"At the same time, extending the atmospheric spin-up period (6h) can also resolve the issue, because simulation results are expected to less susceptible to a parent reanalysis data." (P. 17, L. 636-637)

pg 10, section 4.2 : I do not see the interest of showing here the ability of SMAP only to simulate a single wind event. Outputs from JRA-55 should be added in the comparison to show the interest of SMAP in respect to JRA-55. MARv3.5.2 (at a resolution of 20km) forced by JRA55 underestimates also this event by a factor of 10-15m/s. The interest of using a non-hydrostatic model at 5 km should be highlighted here.

Thank you for the comment. We have included 10m wind speed data from JRA-55 in Fig. 4 and added the following discussion:
"In the figure, 10m wind speed from the parent JRA-55 reanalysis with a horizontal resolution of TL319 (~55 km) is depicted together. Clearly, JRA-55 could not reproduce the strong wind event and an advantage of a high-resolution non-hydrostatic atmospheric model is successfully demonstrated." (P. 11, L. 401-403)

In connection with this point, we thought horizontal resolution of JRA-55 should be mentioned in Sect. 2.3.2: "Dynamical downscaling of atmospheric field from reanalysis data with JMA-NHM". Therefore, it has been described in Sect.2.3.2 as follows:
"Horizontal resolution of JRA-55 is TL319 (~55 km)." (P. 7, L. 240)

pg 12, lines 409-423: the fact that SMAP overestimates surface temperature but underestimates both LWD/SWD fluxes suggests that SMAP is likely too dependent of the forcing data. What about the

latent and sensible heat fluxes? The authors suggests that near-surface snow density is likely too high. I am very sceptic about this explanation. The sensibility of the results to the near-surface snow density can be tested offline. For me, the problem comes from the JRA-55 fields which are too warm and which are used every day to reinitialise the SMAP atmospheric fields.

Thank you for the insightful comment. First of all, regarding the underestimation of downward longwave radiant flux, we think that observation data also has error that affects model evaluation significantly. At the end of Sect. 4.3, we have added the following discussion:
"On the other hand, observation data for downward longwave radiant flux can also have error especially during the winter period due to riming, which may act to increase measured values. In SIGMA-A, measured 2m air temperature often decreased to about –40 °C during the 2013-2014 winter (Fig. 3a). Although such reductions in 2m air temperature during March and April 2014 were followed by significant reductions in downward longwave radiant flux (Fig. 3e), they did not synchronize in December 2013 and January 2014. These results suggest that observed downward longwave radiant flux especially during December 2013 and January 2014 were affected by riming and forced to increase. A reliable quality control technique for automatic downward longwave radiant flux measurements in the polar region should be developed in the future to perform not only model validation but also climate monitoring accurately." (P. 12, L. 443 – P. 13, L. 452)

In the summary and conclusions section, an additional summary regarding this issue has been added as follows:
"On the other hand, observation data for downward longwave radiant flux can also have error especially during the winter period due to riming, which might affect the evaluation." (P. 18, L. 651 - 653)

During the revision, we performed additional data quality control for downward longwave radiant flux. What we performed is that rejecting such data as downward and upward longwave radiant fluxes agree exactly. This situation is caused when extreme riming occurs and these two properties are diagnosed only from sensor temperature. However, our discussion was not affected by the reassessment of measurement data.

Based on these, we now agree with the reviewer's point that the problem comes from the JRA-55 fields which are too warm and which are used every day to reinitialize the SMAP atmospheric fields. At the same time, overestimation of relatively low surface wind speeds (Sect. 4.2) might affect this problem, because it acts to increase sensible heat flux. As a result, we have revised the sentence as follows:

"One possible cause of the model's overestimation of surface temperature is overestimation of the near-surface snow density profile, which would increase the conductive heat flux to the surface (see Sect. 4.5)."

----->

"One possible cause of the model's overestimation of surface temperature is overestimation of the surface wind speeds when they are relatively low (see Sect. 4.2), which acts to heat the surface through increases in sensible heat flux. Of course, overestimation of 2m temperature by the model (see Sect. 4.1) especially during winter (November to March) also may contribute to the error." (P. 13, L. 465-468)

Related to this revision, the following description in the summary and conclusions section has been revised as follows:

"A possible cause for this overestimate is overestimation of the near-surface density profile, as suggested by validation of snow surface height changes."

----->

"A possible cause for this overestimate is overestimation of the surface wind speeds when they are relatively low, which acts to heat the surface through increases in sensible heat flux. In addition, overestimation of 2m temperature by the model especially during winter (November to March) also may contribute to the error." (P. 18, L. 657-660)

pg 12, lines 424-439: it is true that MAR overestimates albedo but as it also overestimates SWD. Due to error compensations (as explained in Fettweis et al., 2017), the MAR surface fields are OK. Here, it is strange that SMAP overestimates temperatures but overestimates albedo and underestimates SWD and LWD.

In the original manuscript, we mentioned two possible reasons for the overestimation of albedo by NHM-SMAP as follows:

"Therefore, future models should consider this process as well as the possibility that NHM-SMAP overestimates snowfall during the summer period." (P. 13, L. 483-484 in the revised manuscript)

In the revision process, we conducted additional model sensitivity tests where ice albedo is set to 0.2 following the suggestion by the reviewer, which is detailed below. The results from the sensitivity tests indicate that simulated SMB did not change significantly compared to the control RE setting (Fig. 8). Based on the result, we reached a conclusion that overestimation of surface albedo by NHM-SMAP can be attributed mainly to overestimates snowfall. These results are mentioned in Sect. 4.7 entitled as "Surface mass balance", and they can also be found in this answer file (our

answer to "pg 14, line 513:").

Pg 13, section 4.6 : the comparison with the melt extent is excellent! Adding here a 2D comparison (nbr of melt days in 2012 for example) should be interesting to evaluate if this agreement is also OK locally. The simulated total melt extent could be good for bad reasons and local overestimation/underestimation of melt can be compensated.

Thank you very much for the encouraging comment. In Fig. S1 of the supplementary file, we have added the 2D comparison figure. At the end of Sect. 4.6, we have added the following explanation regarding the figure:
"Figure S1, which shows observed and simulated total numbers of surface melt days in 2012, supports this argument." (P. 14, L. 520 - 521)

pg 13, line 479: A 2D comparison with other RCM based estimations (RACMO, MAR, ...) is needed here for me. The raw 20km MARv3.5.2 daily outputs forced by JRA55 are available here: ftp://ftp.climato.be/fettweis/MARv3.5.2/Greenland/JRA-55_20km/
and could be used in this paper just by citing Fettweis et al. (2017).

Thank you for the suggestion and providing the data. We considered whether we should use other RCM based estimations or not, and decided to include simulation results by MAR v3.5.2 forced by the same reanalysis data JRA-55 as used in the present study. At the beginning of Sect. 4.7, we have indicated it as follows:

"In addition, simulated SMB data from MAR v3.5.2 forced by JRA-55 (Fettweis et al., 2017) were employed as reference information." (P. 15, L. 525-527)

At present, there are many different points in model formulations and configurations of MAR and NHM-SMAP, namely, resolution, ice sheet mask, dynamic core of atmospheric model, albedo model, water percolation scheme for snow/firn, etc. Therefore, detailed model inter-comparison should be beyond the scope of this paper; however, we do hope to perform such a comparison in the near future.

MAJOR: pg 14, line 507: MAR at 20km is generally able to resolve the ablation zone. The 5 km resolution used here is not an issue here to explain the systematic SMB overestimation in the ablation zone by SMAP. RACMO at 11km works also already very well. Significant biases in energy balance fluxes could explain the underestimation of ablation.

Thank you for the comment. We think the reason why MAR at 20km successfully resolves the ablation area is the introduction of a sub-grid mask, which is not considered by the present version of NHM-SMAP. Based on this consideration, we added the following discussion:

"On the other hand, MAR v3.5.2 with a horizontal resolution of 20km is generally able to resolve the ablation zone well (Fettweis et al., 2017). A possible cause for this success can be attributed to the introduction of sub-grid mask, which is not employed by NHM-SMAP. It appears that statistical downscaling or further dynamical downscaling or introduction of sub-grid mask is inevitable to obtain more realistic SMB estimates." (P. 16, L. 582-586)

Also, in the final section, we have mentioned it again as follows:

"Moreover, statistical downscaling or further dynamical downscaling to a higher spatial resolution than used here, e.g. 1 km (Noel et al. 2016, Wilton et al. 2017) or introduction of sub-grid mask (Fettweis et al., 2017) may inevitably be required to improve the SMB estimates." (P. 18, L. 673 - 675)

pg 14, line 513: to test the problem of the overestimation of albedo in SMAP, an offline simulation using a bare ice albedo of 0.2 could be carried out here and results should be shown in Fig 8.

It is a very nice suggestion. We have performed the suggested model sensitivity tests and discussed the results as follows:

"According to the PROMICE data in the ablation area, ice albedo often decreases to around 0.2 during summer. Therefore, additional model sensitivity tests, where ice albedo is set to 0.2, were performed. Obtained results indicate that simulated SMB did not change significantly compared to the control Richards equation setting (Fig. 8), suggesting that overestimation of surface albedo by NHM-SMAP can be attributed mainly to overestimates snowfall as pointed out in Sect. 4.4." (P. 16, L. 571 - 576)

In accordance with this, Fig. 8 has been updated. In the original manuscript, we did not refer Fig. 8 explicitly, therefore, it has been referred at the beginning of Sect. 4.7 as follows:

"During the study period, 55 measurements were available, and comparison results are presented in Fig. 8." (P. 14, L. 524 – P. 15, L. 525)

Accordingly, the following sentence in the original manuscript (P. 14, L. 512-513) has been removed:

"Moreover, it is imperative that we develop a realistic albedo model for high-density firn and ice that

incorporates the effects of cryoconite."

Also, the following sentence in the original manuscript (P. 16, L. 592-594) has been removed as well:

"This finding underscores the need to develop a realistic albedo model for high-density firn and ice that allows us to consider the effects of darkening of the GrIS by cryoconite and so on.", and the following sentence has been added in the revised manuscript instead:

"It was attributed to overestimation of snowfall." (P. 18, L. 662)

pg 14, line 522, explicit comparison with MAR or RACMO is needed here for me. RACMO or MAR time series could be added in Fig 9.

As mentioned above, we have included simulation results by MAR v3.5.2 forced by JRA-55. In the revised manuscript, we have compared the data with the NHM-SMAP-simulated GrIS SMB in Fig. 10a. The related description are as follows:

"According to simulation results by MAR v3.5.2 forced by JRA-55 (Fettweis et al., 2017) that uses the bucket schemes with irreducible water contents of 8 %, the GrIS SMB during the 2011-2012 mass balance year was relatively low (147 Gt year$^{-1}$), then increased greatly in 2012-2013 (473 Gt year$^{-1}$) and decreased slightly in 2013-2014 (403 Gt year$^{-1}$). Our model, which tends to simulate lower SMB compared to MAR v3.5.2, produced a similar sequence in those years, with accumulated SMBs at the end of each mass balance year of –23, 420, and 312 Gt year$^{-1}$, respectively (Fig. 10a)." (P. 16, L. 591 - 596)

pg 15, lines 532-540: such sensitivity to the irreducible water content is also simulated by MAR which uses a value of 8%.

Thank you for the information. The provided information has been included in the revised manuscript as mentioned in the previous answer.

---

## Author Comment (AC4) · 22 Sep 2017

**Reply to Leo van Kampenhout**

We sincerely appreciate Leo van Kampenhout for taking the time to review our paper. Below we describe our responses (in blue text) point-by-point to each comment (in black text). In addition, we indicate revisions in the updated manuscript together with new line numbers. Please also refer to the revised marked-up manuscript uploaded in the discussion board.

I agree with Xavier Fettweis that this work would be a welcome addition to the literature and the wider RCM modelling community. Some questions came up while reading the manuscript, in particular about the spinup method and the effect of percolation.

Thank you for the comment. We agree with the reviewer's point that we should detail more about the model spin-up and the effect of percolation. Please check our answers below.

L 238-240: I searched Dumont et al. (2014) for their spin-up procedure, but failed to find information on this. Did the authors obtain the method details through personal communication?

I am sorry the original description was old and incorrect. We have revised the sentence as follows:
"The initial snow/firn/ice physical conditions for the entire GrIS on 1 September 2011 were prepared by performing a 30year spin-up of the NHM-SMAP model following the procedure of Dumont et al. (2014)."
----->
"The initial top 30 m snow/firn/ice physical conditions for the entire GrIS on 1 September 2011 were prepared by performing a 30year spin-up of the NHM-SMAP model." (P. 7, L. 260-261)

L 238-240: I was wondering whether 30 years is sufficient to get a 30-m snowpack into equilibrium with the climate. Was there any remaining drift in e.g. the bottom layer temperature? What climate years were used to forced the spinup?

First, before starting model spin-up, we attempted prepare realistic initial profiles for snow/firn/ice physical conditions in the GrIS as much as possible. Thanks to this, we did not encounter any drift at the beginning of model spin-up. The performed procedure to prepare the initial conditions before the model spin-up has been described in the revised manuscript as follows:
"Before performing the model spin-up, the initial profiles for snow/firn/ice physical conditions in the GrIS were given following the procedure presented by Lefebre et al. (2005) and

properties for snow/firn microstructure (e.g., optically equivalent grain size and grain shape) were given from the firn core analysis at SIGMA-A (Yamaguchi et al., 2014) equally in the GrIS." (P. 7, L. 261 – P. 8, L. 265)

As for climate years used to force the model spin-up, we used the data during the 2010-2011 mass balance year. First, we performed JMA-NHM stand-alone simulations forced by JRA-55 during the period. Then, the simulation results for surface atmospheric conditions forced SMAP 30 times cyclically (off-line calculation). Of course, it is ideal to perform continuous (not cyclic) 30year spin-up; however, it was not realistic due to computational costs. In the revised manuscript, it is described as follows:
"From the state, surface atmospheric conditions from September 2010 to August 2011 simulated by JMA-NHM forced by JRA-55 were used to drive SMAP for 30 times cyclically." (P. 8, L. 265-267)

Reference:
Lefebre, F., Fettweis, X., Gallée, H., Van Ypersele, J.-P., Marbaix, P., Greuell, W., Calanca, P.: Evaluation of a high-resolution regional climate simulation over Greenland, Climate Dynamics, 25, 99, doi:10.1007/s00382-005-0005-8, 2005.

L 242: It reads like you started with zero snow depth at the beginning of the spinup period. The zero heat flux is then assumed at the bottom of the snow pack, not at 30 m, which is almost never reached? (which you mention in 245-246)

We "did not" start with zero snow depth at the beginning of the spin-up period as mentioned above. During the simulation period, the thickness of snow/firn/ice is always constant: 30 m. It is mentioned in the revised manuscript as follows:
"The thickness of snow/firn/ice is always set to constant (30 m) during the calculation. In case snow accumulation or ablation is simulated, the thickness of the bottom model layer is modified accordingly." (P. 7, L. 258 – L. 260)

L484-485: Figure 10 shows that runoff is larger for larger IWC value (6%), so the "piping" effect must be dominated by something else. Otherwise, the 2%-bucket model would have produced the largest runoff value.

We agree with the reviewer's point that 2%-bucket model setting can heat snow/firn effectively, then result in earlier onset of melting, which can produce large runoff. However, in the

sensitivity tests, we did not consider feedbacks that have more than a year time-scale due to our test setting. In the revised manuscript, we have added the following explanations regarding the setting of model sensitivity tests that changed water percolation schemes:

"In the sensitivity tests, profiles for snow/firn/ice physical conditions were reset at the beginning of the 2011–2012, 2012–2013, and 2013–2014 mass balance years by referring to the simulation data from the on-line version of NHM-SMAP. It means that feedbacks, which have more than a year time-scale, are not considered." (P. 15, L. 539-542)

L 497-502: The authors do not supply any proof of their statement that the formation of ice layers is the reason for the increased runoff. In particular, they do not present melt and refreezing as separate terms. After the formation of (sub-surface) ice layers, one expects the melt to stay roughly the same order of magnitude, yet see a drop in refreezing due to the added effect of lateral runoff.

On the other hand, an increase in runoff could also occur due to increased melt. The reasoning is that when you have higher IWC and more refreezing, warmer snow will result which leads to stronger metamorphism and larger grains that lower the albedo. The warm snow also will persist throughout winter and helps to bring snow to the melting point in spring. This behaviour is also seen in other models. It would benefit this paper if light could be shed on the exact processes that are dominant in this study.

Thank you for the insightful comments and suggestion. Following the suggestion, we have included a figure showing melt and refreeze rates, which are monitored in NHM-SMAP operationally. In the revised manuscript, it is discussed as follows:

"To confirm the discussion, the GrIS-area-integrated daily melt and refreeze rates were investigated (Fig. 9). In the figure, results for the 2011-2012 mass balance year are shown, whereas results for other mass balance years are depicted in Fig. S3. During the 2011-2012 mass balance year, simulated daily melt rates were almost the same among the results from Richards equation scheme and two bucket schemes (Fig. 9a); however, refreeze rates from the control Richards equation scheme were much lower compared to other results (Fig. 9b), which is an evidence for the above-mentioned more impermeable ice in the results from Richards equation scheme. The same characteristics could be found in other mass balance years (Fig. S3)." (P. 15, L. 554-561)

What we found are basically the same as the reviewer's recognition.

---

## Author Comment (AC6) · 22 Sep 2017

**Table S1. Model performance in terms of simulating hourly 2 m water vapor pressure (in hPa) at each AWS on the GrIS (Figure 1). Note that the evaluation were conducted at only SIGMA and PROMICE sites. ME, RMSE, and $R^2$ are the mean error (the average of the difference between simulated values and observed values), and the coefficient of determination, respectively. Number of observations (OBS) employed for the comparison are also listed.**

| Sites | ME (hPa) | RMSE (hPa) | $R^2$ | Number of observations |
|---|---|---|---|---|
| SIGMA-A | 0.07 | 0.36 | 0.95 | 18998 |
| SIGMA-B | 0.21 | 0.48 | 0.94 | 18541 |
| KPC_U | –0.01 | 0.44 | 0.95 | 26139 |
| SCO_U | –0.16 | 0.62 | 0.90 | 25786 |
| TAS_U | –0.33 | 0.76 | 0.84 | 23263 |
| QAS_L | –0.53 | 0.88 | 0.89 | 23483 |
| QAS_A | –0.42 | 0.77 | 0.89 | 8678 |
| NUK_L | –0.23 | 0.67 | 0.92 | 21933 |
| NUK_U | –0.30 | 0.63 | 0.92 | 20908 |
| NUK_N | –0.23 | 0.56 | 0.93 | 19955 |
| KAN_L | –0.02 | 0.52 | 0.94 | 25518 |
| KAN_M | –0.15 | 0.59 | 0.92 | 20379 |
| KAN_U | –0.05 | 0.46 | 0.93 | 22925 |
| UPE_L | –0.27 | 0.69 | 0.92 | 25409 |
| UPE_U | –0.27 | 0.56 | 0.95 | 23036 |
| Mean value | –0.18 | 0.60 | 0.92 | |

**Table S2. Model performance in terms of simulating hourly surface pressure (in hPa) at each AWS on the GrIS (Figure 1). Elevation differences between the reality and NHM-SMAP are indicated together.**

| Sites | ME (hPa) | RMSE (hPa) | $R^2$ | Number of observations | Elevation difference (m) |
|---|---|---|---|---|---|
| SIGMA-A | –2.8 | 2.9 | 0.99 | 18998 | 4 |
| SIGMA-B | 17.4 | 17.4 | 0.99 | 18550 | –165 |
| Summit | –7.6 | 8.9 | 0.86 | 13064 | 44 |
| S-Dome | –4.3 | 4.4 | 1.00 | 11161 | 20 |
| KPC_U | –5.5 | 5.6 | 0.99 | 26304 | 23 |
| SCO_U | –23.1 | 23.2 | 0.98 | 26249 | 176 |
| TAS_U | –2.3 | 2.6 | 0.99 | 23330 | 1 |
| QAS_L | –12.5 | 12.6 | 0.99 | 26302 | 85 |
| QAS_A | –13.9 | 13.9 | 1.00 | 9267 | 104 |
| NUK_L | –7.5 | 7.6 | 0.99 | 26296 | 26 |
| NUK_U | –13.0 | 13.1 | 0.98 | 20933 | 85 |
| NUK_N | –8.4 | 8.5 | 0.99 | 23570 | 46 |
| KAN_L | 5.6 | 5.7 | 0.99 | 26303 | –74 |
| KAN_M | –7.8 | 8.0 | 0.98 | 21208 | 49 |
| KAN_U | –3.7 | 3.7 | 0.99 | 24084 | 20 |
| UPE_L | –7.2 | 7.3 | 0.98 | 25743 | 34 |
| UPE_U | –8.6 | 8.7 | 0.99 | 26300 | 77 |
| Mean value | –8.0 | 8.9 | 0.98 | | |

**Table S3. Model performance in terms of simulating hourly 10 m wind speed (in m s$^{-1}$) at each AWS on the GrIS (Figure 1).**

| Sites | ME (m s$^{-1}$) | RMSE (m s$^{-1}$) | R$^2$ | Number of observations |
|---|---|---|---|---|
| SIGMA-A | –0.5 | 2.6 | 0.40 | 17846 |
| SIGMA-B | 1.0 | 3.2 | 0.14 | 17851 |
| Summit | –0.7 | 2.5 | 0.54 | 18825 |
| S-Dome | –2.0 | 4.0 | 0.76 | 10624 |
| KPC_U | 0.4 | 1.7 | 0.65 | 25921 |
| SCO_U | –0.2 | 2.3 | 0.13 | 25774 |
| TAS_U | 2.5 | 4.3 | 0.68 | 22977 |
| QAS_L | 0.2 | 2.8 | 0.51 | 23423 |
| QAS_A | –0.6 | 2.5 | 0.59 | 8481 |
| NUK_L | 0.4 | 2.3 | 0.52 | 21808 |
| NUK_U | 2.2 | 3.2 | 0.64 | 20807 |
| NUK_N | –0.3 | 2.4 | 0.65 | 19773 |
| KAN_L | 0.8 | 2.4 | 0.54 | 25432 |
| KAN_M | –0.1 | 2.3 | 0.72 | 21047 |
| KAN_U | –1.4 | 2.8 | 0.78 | 22660 |
| UPE_L | 1.3 | 3.1 | 0.44 | 25051 |
| UPE_U | 0.6 | 2.5 | 0.69 | 22906 |
| Mean value | 0.2 | 2.7 | 0.55 | |

**Table S4. Model performance in terms of simulating hourly downward shortwave radiant flux (in W m$^{-2}$) at each AWS on the GrIS (Figure 1).**

| Sites | ME (W m$^{-2}$) | RMSE (W m$^{-2}$) | R$^2$ | Number of observations |
|---|---|---|---|---|
| SIGMA-A | −13.5 | 60.2 | 0.86 | 8077 |
| SIGMA-B | −9.4 | 72.6 | 0.80 | 8069 |
| Summit | −9.1 | 75.9 | 0.88 | 10945 |
| S-Dome | 52.6 | 112.3 | 0.82 | 10556 |
| KPC_U | −28.6 | 56.0 | 0.90 | 11443 |
| SCO_U | 0.6 | 69.0 | 0.88 | 10972 |
| TAS_U | −9.6 | 88.9 | 0.81 | 8588 |
| QAS_L | 16.6 | 96.5 | 0.83 | 11229 |
| QAS_A | −3.8 | 103.7 | 0.81 | 3962 |
| NUK_L | 2.2 | 90.8 | 0.83 | 8384 |
| NUK_U | −10.5 | 82.8 | 0.87 | 8341 |
| NUK_N | 4.4 | 84.5 | 0.86 | 9534 |
| KAN_L | −17.1 | 127.3 | 0.70 | 10837 |
| KAN_M | −16.4 | 73.0 | 0.88 | 8510 |
| KAN_U | −39.4 | 81.3 | 0.91 | 10467 |
| UPE_L | −0.7 | 78.5 | 0.83 | 11007 |
| UPE_U | −7.0 | 65.0 | 0.88 | 11061 |
| Mean value | −5.2 | 83.4 | 0.84 | |

**Table S5. Model performance in terms of simulating hourly downward longwave radiant flux (in W m$^{-2}$) at each AWS on the GrIS (Figure 1). Note that the evaluation were conducted at only SIGMA and PROMICE sites.**

| Sites | ME (W m$^{-2}$) | RMSE (W m$^{-2}$) | R$^2$ | Number of observations |
|---|---|---|---|---|
| SIGMA-A | −24.3 | 36.6 | 0.71 | 18353 |
| SIGMA-B | −14.4 | 31.6 | 0.72 | 18440 |
| KPC_U | −14.3 | 28.3 | 0.74 | 26066 |
| SCO_U | −17.0 | 28.3 | 0.78 | 26221 |
| TAS_U | −20.5 | 32.7 | 0.66 | 23107 |
| QAS_L | −19.8 | 30.2 | 0.80 | 26216 |
| QAS_A | −21.4 | 32.5 | 0.76 | 9209 |
| NUK_L | −21.7 | 32.0 | 0.80 | 21835 |
| NUK_U | −13.6 | 28.6 | 0.78 | 20827 |
| NUK_N | −21.3 | 15.0 | 0.77 | 23441 |
| KAN_L | −13.0 | 28.1 | 0.76 | 26155 |
| KAN_M | −10.7 | 28.5 | 0.75 | 21140 |
| KAN_U | −11.7 | 29.8 | 0.71 | 23962 |
| UPE_L | −22.2 | 35.8 | 0.72 | 25562 |
| UPE_U | −13.9 | 29.8 | 0.77 | 26225 |
| Mean value | −17.3 | 29.9 | 0.75 | |

**Table S6. Model performance in terms of simulating hourly snow/firn/ice surface temperature (in ºC) at each AWS on the GrIS (Figure 1). Note that the evaluation were conducted at only SIGMA and PROMICE sites.**

| Sites | ME (°C) | RMSE (°C) | $R^2$ | Number of observations |
|---|---|---|---|---|
| SIGMA-A | 2.3 | 4.7 | 0.91 | 19007 |
| SIGMA-B | 3.2 | 4.9 | 0.91 | 18551 |
| KPC_U | 2.6 | 4.8 | 0.93 | 26139 |
| SCO_U | 1.1 | 4.3 | 0.82 | 26235 |
| TAS_U | 1.7 | 3.2 | 0.82 | 23316 |
| QAS_L | 0.4 | 2.2 | 0.87 | 26301 |
| QAS_A | 0.0 | 2.6 | 0.90 | 9264 |
| NUK_L | 0.4 | 2.7 | 0.88 | 21944 |
| NUK_U | –0.3 | 2.7 | 0.90 | 20920 |
| NUK_N | 0.1 | 2.8 | 0.89 | 22793 |
| KAN_L | 1.1 | 3.2 | 0.90 | 26284 |
| KAN_M | 1.0 | 3.5 | 0.91 | 21184 |
| KAN_U | 0.9 | 3.3 | 0.93 | 24039 |
| UPE_L | 2.0 | 4.5 | 0.85 | 25747 |
| UPE_U | 1.0 | 3.3 | 0.92 | 26291 |
| Mean value | 1.2 | 3.5 | 0.89 | |

**Table S7. Model performance in terms of simulating hourly snow and ice albedo at each AWS on the GrIS (Figure 1). Note that the evaluation were conducted at only SIGMA and PROMICE sites.**

| Sites | ME | RMSE | $R^2$ | Number of observations |
|---|---|---|---|---|
| SIGMA-A | 0.02 | 0.07 | 0.04 | 3150 |
| SIGMA-B | 0.07 | 0.15 | 0.06 | 3250 |
| KPC_U | 0.09 | 0.13 | 0.06 | 4451 |
| SCO_U | 0.22 | 0.27 | 0.09 | 5297 |
| TAS_U | 0.15 | 0.24 | 0.10 | 3627 |
| QAS_L | 0.32 | 0.41 | 0.12 | 6415 |
| QAS_A | 0.15 | 0.25 | 0.03 | 2252 |
| NUK_L | 0.27 | 0.32 | 0.13 | 4501 |
| NUK_U | 0.20 | 0.25 | 0.09 | 4752 |
| NUK_N | 0.23 | 0.33 | 0.12 | 5352 |
| KAN_L | 0.19 | 0.23 | 0.16 | 6003 |
| KAN_M | 0.17 | 0.25 | 0.12 | 4571 |
| KAN_U | 0.08 | 0.11 | 0.07 | 5967 |
| UPE_L | 0.11 | 0.17 | 0.19 | 5136 |
| UPE_U | 0.15 | 0.22 | 0.10 | 5243 |
| Men value | 0.16 | 0.23 | 0.10 | |

**Table S8. Model performance (on-line version of NHM-SMAP) in terms of simulating JJA hourly 2m air temperature at each AWS on the GrIS (Figure 1).**

| Sites | ME (°C) | RMSE (°C) | $R^2$ | Number of observations | Elevation (m) |
|---|---|---|---|---|---|
| SIGMA-A | −0.1 | 1.5 | 0.83 | 5894 | 1490 |
| SIGMA-B | 1.1 | 1.7 | 0.87 | 5446 | 944 |
| Summit | −0.1 | 3.5 | 0.67 | 5772 | 3208 |
| S-Dome | 0.2 | 2.2 | 0.80 | 4521 | 2901 |
| KPC_U | −1.2 | 2.0 | 0.79 | 6624 | 870 |
| SCO_U | −1.7 | 2.6 | 0.57 | 6122 | 980 |
| TAS_U | 2.6 | 3.2 | 0.41 | 4414 | 570 |
| QAS_L | 1.4 | 2.3 | 0.45 | 4273 | 290 |
| QAS_A | −1.4 | 2.3 | 0.48 | 1992 | 1010 |
| NUK_L | 0.1 | 1.7 | 0.53 | 5351 | 550 |
| NUK_U | −0.7 | 2.0 | 0.67 | 5308 | 1130 |
| NUK_N | −0.3 | 1.5 | 0.74 | 3227 | 920 |
| KAN_L | 0.1 | 1.2 | 0.76 | 5960 | 680 |
| KAN_M | −0.8 | 1.9 | 0.80 | 5097 | 1270 |
| KAN_U | −1.5 | 2.5 | 0.81 | 6618 | 1840 |
| UPE_L | −0.1 | 1.7 | 0.62 | 6360 | 220 |
| UPE_U | −0.6 | 1.4 | 0.87 | 5044 | 940 |
| Mean value | −0.2 | 2.1 | 0.69 | | |

[Figure]

**Figure S1: (a) Observed and (b) simulated number of the GrIS surface melt days in 2012. Observation data are from Mote (2014).**

[Figure]

**Figure S2: The NHM-SMAP simulated accumulated GrIS SMB (in mm) during the (a) 2011-2012, (b) 2012-2013, and (c) 2013-2014 mass balance years (September to August).**

[Figure]

**Figure S3: Sensitivity to the choice of vertical water movement scheme of the simulated top 30m integrated (a and c) melt and (b and d) refreeze for the GrIS during the (a and b) 2012-2013 and (c and d) 2013-2014 mass balance years. RE indicates the default setting for vertical water movement in snow and firn based on the Richards equation; Bucket_6% and Bucket_2% are alternative settings based on simple bucket schemes with irreducible water contents of 6 % and 2 % of the pore volume.**

---

## Author Comment (AC7) · 22 Sep 2017

Sorry, there is a typo in our response to the comment for L238-240:

Thanks to this, we did not encounter any drift at the beginning of model spin-up.

——>

Thanks to this, we did not encounter any drift at the end of model spin-up.

---

## Author Response (AR2)

January 9, 2018
Dr. Masashi Niwano
Meteorological Research Institute
Japan Meteorological Agency
Tsukuba 305-0052, Japan
Phone: +81-29-853-8714
FAX: +81-29-855-6936
e-mail: mniwano@mri-jma.go.jp

Prof. Tedesco
Editor, The Cryosphere

Dear Prof. Tedesco:

    We are happy to know that our revised manuscript entitled as "NHM-SMAP: Spatially and temporally high resolution non-hydrostatic atmospheric model coupled with detailed snow process model for Greenland Ice Sheet" by Masashi Niwano et al. [Paper # tc-2017-115] submitted to the journal The Cryosphere has been accepted. Thank you again for taking the time to check our manuscript and obtaining expert reviewers.

    In response to the Editor's comment: "Please, consider using a more quantitative term on line 28, in the abstract, rather than using the qualitative word 'well'.", we have revised the sentence as follows:

"In particular, it reproduced the GrIS surface melt area extent well."

---->

"In particular, it successfully reproduced the temporal evolution of the GrIS surface melt area extent, as well as the record melt event around 12 July 2012, at which time the simulated melt area extent reached 92.4 %."

Yours sincerely,
Masashi Niwano and co-authors